# Causal Effect Estimation with Mixed Latent Confounders and Post-treatment Variables

**Yaochen Zhu[1], Jing Ma[2], Liang Wu[3], Qi Guo[3], Liangjie Hong[3], Jundong Li[1]**
[1]University of Virginia, [2]Case Western Reserve University, [3]LinkedIn Inc.
`{uqp4qh, jundong}@virginia.edu`
`jing.ma5@case.edu`
`{liawu, qguo, liahong}@linkedin.com`

## Abstract

Causal inference from observational data has attracted considerable attention among researchers. One main obstacle is the handling of confounders. As direct measurement of confounders may not be feasible, recent methods seek to address the confounding bias via proxy variables, i.e., covariates postulated to be conducive to the inference of latent confounders. However, the selected proxies may scramble both confounders and post-treatment variables in practice, which risks biasing the estimation by controlling for variables affected by the treatment. In this paper, we systematically investigate the bias due to latent post-treatment variables, i.e., *latent post-treatment bias*, in causal effect estimation. Specifically, we first derive the bias when selected proxies scramble both latent confounders and post-treatment variables, which we demonstrate can be arbitrarily bad. We then propose a Confounder-identifiable VAE (CiVAE) to address the bias. Based on a mild assumption that the prior of latent variables that generate the proxy belongs to a general exponential family with at least one invertible sufficient statistic in the factorized part, CiVAE *individually* identifies latent confounders and latent post-treatment variables up to bijective transformations. We then prove that with individual identification, the intractable disentanglement problem of latent confounders and post-treatment variables can be transformed into a tractable independence test problem despite arbitrary dependence may exist among them. Finally, we prove that the true causal effects can be unbiasedly estimated with transformed confounders inferred by CiVAE. Experiments on both simulated and real-world datasets demonstrate significantly improved robustness of CiVAE.

## 1 Introduction

Causal inference, which aims to infer cause-and-effect relations from data, has gained increasing prominence in various fields, such as social science, economics, and public health (Glass et al., 2013; Johansson et al., 2016; Prosperi et al., 2020). Traditional methods rely on the gold standard of randomized control trials (RCT) to draw causal conclusions via experimentation (Cook et al., 2002). Recently, more attention has been dedicated to causal inference from observational data, where treatments, outcomes, and unit covariates are passively observed, and researchers have no control over the treatment assignment mechanism (Shalit et al., 2017; Shi et al., 2019; Wager & Athey, 2018).

One main obstacle to inferring causal relations from observational data is confounding bias, which occurs when we fail to account for the systematic difference between treatment and non-treatment groups due to variables that causally influence treatment and outcome, i.e., unobserved confounders (Zhu et al., 2024; Jager et al., 2008). If confounders can be measured, we can address the bias by controlling them via covariate adjustment (Pocock et al., 2002) or propensity score re-weighting (Li et al., 2018). However, confounders are not always measurable (Kuroki & Pearl, 2014). Therefore, recent efforts have been devoted to adjusting for unobserved confounders based on their proxies, which are observed covariates postulated to be causally related with the unobserved confounders (Miao et al., 2018; Yao et al., 2018; Madras et al., 2019). One exemplar work is the causal effect variational auto-encoder (CEVAE) (Louizos et al., 2017), which has demonstrated that confounding bias can be mitigated by controlling latent variables inferred from the proxies of confounders.

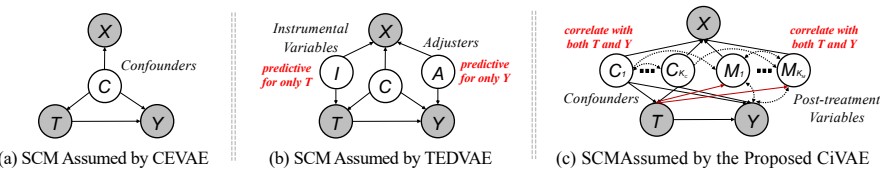

(a) SCM Assumed by CEVAE    (b) SCM Assumed by TEDVAE    (c) SCMAssumed by the Proposed CiVAE

Figure 2: Comparison between the causal models assumed by CEVAE, TEDVAE, and CiVAE. Bi-directional dashed arrow means we allow arbitrary correlation between the two variables

Although proxy-based methods have achieved substantial progress in recent years, they may risk controlling latent post-treatment variables scrambled in the proxies, where **latent post-treatment bias** can be introduced in the estimation. The negative effects of controlling *observed* post-treatment variables have been investigated in prior research (Acharya et al., 2016; Elwert & Winship, 2014; King & Zeng, 2006). For example, Montgomery et al. (2018) found that more than 50% of the papers published in top journals of politics *inadvertently controlled post-treatment variables* in the experimental setting, even though the researchers had complete control over which covariates to control for. On this basis, we postulate that the post-treatment bias could be even worse for proxy-based methods in the setting of observational study where available covariates are passively recorded. In addition, the post-treatment variables can be **latent** and scrambled into the observed covariates together with the latent confounders, which makes them difficult to disentangle.

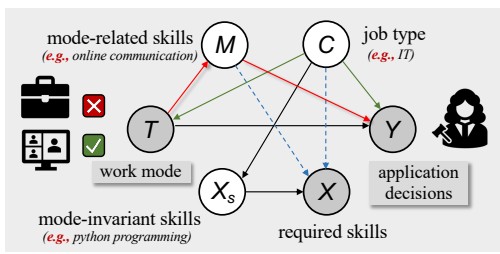

Figure 1: Example of latent post-treatment bias.

Consider a real-world example from LinkedIn. We found that *changing* a job from onsite to online mode causes applicants to make different decisions, and we want to estimate the causal effects of *switching a job from onsite to online mode* on *the decisions of the applicants* (Here, the units under study are the jobs instead of applicants). To study the case, we collected two groups of online (treated) and onsite (control) jobs, where (the average) application decisions of job seekers are treated as the outcome. Clearly, job type is a confounder as certain jobs (e.g., IT) are more likely to permit online work and are more difficult to apply. When job type is not provided in the posting, the required skills of the job (which is mandatory) can be used as the proxy of the confounder "job type". However, **a caveat** is that switching to an online work mode may also alter the required skills of a job, thereby affecting the qualification and, therefore, the decision of the applicants. Consequently, directly using the "required skills" as the proxy of the confounder "job type" for adjustment could unintentionally control latent mediators (changed skills), which introduces latent post-treatment bias in the causal effect estimation.

Similar examples are ubiquitous in observational studies in other fields where data are collected *post hoc* and pre-treatment and post-treatment components entangle in the observed covariates. For example, for medical treatment effect estimation, a common circumstance is that only post-treatment test results of the patients are available (Liu et al., 2012). Some results, such as the changes in blood pressure, could be influenced by the treatment, while others (depending on the specific treatment) could remain invariant to the treatment but are indicative of a patient's pre-treatment health (i.e., an important confounder for treatment effect estimation). The problem can be exacerbated in politics and economy (Klar et al., 2020). For example, when estimating the causal effect of historical policies, usually only yearly-aggregated data on the covariates, such as GDP, employment rates, or social welfare are available. These variables are often a mixture of pre-policy conditions (e.g., GDP from the beginning of the year to the date the policy took effect) reflecting the latent pre-policy state of the area (i.e., confounders) and post-policy outcomes (e.g., GDP after the policy), where latent post-treatment bias can be introduced if these covariates are treated as confounder proxies (Homola et al., 2024).

Addressing the **latent post-treatment bias** faces multi-faceted challenges. First, there lacks a theoretical formulation of the bias when proxies scramble latent post-treatment variables for existing proxy-based methods. In addition, it is difficult to distinguish confounders and post-treatment variables in the latent space due to their similar effects on observed covariates. Existing covariate disentanglement-based methods, e.g., TEDVAE (Zhang et al., 2021), focus on an easier task of disentangling latent confounders with adjusters and instrumental variables, which can be achieved by

leveraging their different predictive abilities w.r.t. the treatment and outcome. However, since both latent confounders and post-treatment variables correlate with the treatment and the outcome, they cannot be disentangled by these methods. Finally, even if latent confounders can be distinguished from post-treatment variables, since most existing latent variable models have no identifiability guarantee (Khemakhem et al., 2020), it is unclear whether controlling the inferred latent variables, which may be arbitrary transformations of the true confounders, can provide unbiased estimations.

To address the challenges, we first analyze existing proxy-based methods when selected proxies scramble both latent confounders and latent post-treatment variables, where we show the estimation can be arbitrarily biased. We then propose a Confounder-identifiable VAE (CiVAE) to address the latent post-treatment bias. Specifically, we prove that based on a mild assumption that the prior of latent variables that generate the observed proxy belongs to a general exponential family with at least one invertible sufficient statistic in the factorized part, latent confounders and latent post-treatment variables can be *individually* identified up to *bijective transformations* and permutation. With such an identifiability guarantee, based on the causal relations among confounders, post-treatment variables, and treatment, we further demonstrate that the inferred confounders could be properly disentangled from the latent post-treatment variables with pair-wise conditional independence tests, even if they may have arbitrary dependence with each other. Finally, we prove that the true causal effects can be unbiasedly estimated based on the confounders inferred by CiVAE. Experiments on both simulated and real-world datasets demonstrate the robustness of CiVAE to latent post-treatment bias.

## 2 PROBLEM FORMULATION

In this paper, we assume the causal model in Fig. 2-(c). We use a binary random variable $T$ to denote the treatment, a random vector $\boldsymbol{X} \in \mathbb{R}^{K_X}$ to denote the observed covariates (i.e., the proxy), and a random scalar $Y \in \mathbb{R}$ to denote the outcome. Furthermore, the observed covariates $\boldsymbol{X}$ are assumed to be generated from $K_C$ independent latent confounders $\boldsymbol{C} \triangleq [C_1, C_2..., C_{K_C}]$ causally influencing both $T$ and $Y$, and $K_M$ latent post-treatment variables $\boldsymbol{M} \triangleq [M_1, M_2..., M_{K_M}]$ under the causal influence of the treatment (where the relation between $\boldsymbol{M}$ and $Y$ can be arbitrary). We use the random vector $\boldsymbol{Z} \triangleq [\boldsymbol{C}||\boldsymbol{M}] \in \mathbb{R}^{K_Z=K_C+K_M}$ to denote all relevant latent factors. We assume that beyond $\boldsymbol{C}$, there are no latent confounders that confound the latent variables $\boldsymbol{Z}$. **Our aim** is to estimate the average causal effects of treatment $T$ on outcome $Y$ with the auxiliary confounder information in $\boldsymbol{X}$, where the estimation should be devoid of both confounding bias and post-treatment bias.

## 3 THEORETICAL ANALYSIS OF LATENT POST-TREATMENT BIAS

### 3.1 PRELIMINARIES AND ASSUMPTIONS

To achieve such a purpose, we first define the (conditional) average treatment effects (C/ATE) when covariates $\boldsymbol{X}$ scramble both latent confounders $\boldsymbol{C}$ and (mixed-in) post-treatment variables $\boldsymbol{M}$. We then define the post-treatment bias when covariates $\boldsymbol{X}$ are directly used as the proxy of confounders. To facilitate the analysis, we make the following assumption regarding the causal generative process.

**Assumption 1.** *We assume $\boldsymbol{X} = f(\boldsymbol{C}, \boldsymbol{M}) + \boldsymbol{\epsilon}$, where $f$ is a deterministic function that combines latent confounders $\boldsymbol{C}$ and some mixed-in latent post-treatment variables $\boldsymbol{M}$ into the observations $\boldsymbol{X}$, and $\boldsymbol{\epsilon}$ is a random noise. In addition, we assume that the function $f$ is **injective**[1]; beyond injectivity, $f$ can be arbitrarily nonlinear. We use $f^{\dagger} : \boldsymbol{X} \to [\boldsymbol{C}||\boldsymbol{M}]$ to denote its left inverse. We use $f_C^{\dagger} : \boldsymbol{X} \to \boldsymbol{C}$ and $f_M^{\dagger} : \boldsymbol{X} \to \boldsymbol{M}$ to denote the mapping from $\boldsymbol{X}$ to $\boldsymbol{C}$, $\boldsymbol{M}$, respectively.*

Similar assumptions between $\boldsymbol{X}$ and $\boldsymbol{C}$ are commonly made either explicitly or implicitly in most existing proxy-of-confounder-based methods. For example, if both $\boldsymbol{X}$ and $\boldsymbol{C}$ are categorical, Kuroki & Pearl (2014) assume that $\boldsymbol{X}$ has at least the same number of categories as $\boldsymbol{C}$ and the conditional distribution $p(\boldsymbol{X}|\boldsymbol{C})$ is non-degenerate, whereas the effect restoration algorithm (Miao et al., 2018) assumes that $K_X = K_C$ and the matrix $[p(\boldsymbol{X}_i|\boldsymbol{C}_j)]_{i,j=1,1}^{K_X,K_C}$ to be full-rank. Although CEVAE (Louizos et al., 2017) makes no explicit injectivity assumption between $\boldsymbol{C}$ and $\boldsymbol{X}$, it requires that the joint distribution $p(\boldsymbol{C}, \boldsymbol{X}, T, Y)$ can be fully recovered from the observations $(\boldsymbol{X}, T, Y)$.

---

[1]Note that when $K_X > K_Z$, non-injectives have measure zero in functional space $\{f : \{\boldsymbol{C}, \boldsymbol{M}\} \to \boldsymbol{X}\}$.

Anandkumar et al. (2014) show that some of the possible identification criteria for the recovery of $p(\boldsymbol{C}, \boldsymbol{X}, T, Y)$ include **(i)** $\boldsymbol{X}$ having multiple independent views of $\boldsymbol{C}$ (Edwards et al., 2015), or **(ii)** $\boldsymbol{C}$ being categorical and $\boldsymbol{X}$ being a mixture of Gaussian with mixture components determined by $\boldsymbol{C}$ (that is, $\boldsymbol{X}$ is generated by bijective mapping of $\boldsymbol{C}$ to the mean of the corresponding mixture component with added Gaussian noise).

In the following part, we will temporarily omit the noise $\epsilon$ to gain a better intuition of latent post-treatment bias (but all the conclusions will still hold in the posterior sense). In Section 4, we assume noise exists and demonstrate our method can still properly identify the latent confounders.

## 3.2 CAUSAL ESTIMAND AND THE TRUE ATE

Based on Assumption 1, we are ready to define the estimand of average treatment effect (ATE) through controlling the covariates $\boldsymbol{X}'$, as well as the true (conditional) average treatment effects.

**Definition 1.** *(DCEV & DEV). We define the Difference in Conditional Expected Values (DCEV) as:*

$$DCEV(\boldsymbol{x}') = \mathbb{E}[Y|T=1, \boldsymbol{X}'=\boldsymbol{x}'] - \mathbb{E}[Y|T=0, \boldsymbol{X}'=\boldsymbol{x}'], \tag{1}$$

*which is the difference of the expected value of $Y$ for units with variable $\boldsymbol{X}' = \boldsymbol{x}'$ in the treatment group and the non-treatment group. Based on $DCEV(\boldsymbol{x}')$, we define the Difference in Expected Value (DEV) as $DEV(\boldsymbol{X}') = \mathbb{E}_{p(\boldsymbol{X}')}[DCEV(\boldsymbol{X}')]$ as the expectation of $DCEV$ w.r.t. $p(\boldsymbol{X}')$.*

$DEV(\boldsymbol{X}')$ denotes the estimand of ATE when $\boldsymbol{X}'$ is the covariates that we choose to control (i.e., calculate the expected difference in each stratum of $\boldsymbol{X}' = \boldsymbol{x}'$). If $\boldsymbol{X}' = \emptyset$, $DEV(\emptyset)$ represents the *naive estimator* that directly calculates the expected difference of the outcome $Y$ between the treatment group and the non-treatment group. With the causal estimand $DEV(\boldsymbol{X}')$ defined, we then derive the true causal effects with the covariates $\boldsymbol{X}'$ when it scrambles both latent confounders and post-treatment variables according to the generative process described in Assumption 1:

**Definition 2.** *Under Assumption 1, we define the Conditional Average Treatment Effect (CATE) for individuals with observed covariates $\boldsymbol{X} = \boldsymbol{x}$ by controlling only the confounder part in $\boldsymbol{X}$ as:*

$$CATE(\boldsymbol{x}) = \mathbb{E}[Y|T=1, \boldsymbol{C}=f_C^\dagger(\boldsymbol{x})] - \mathbb{E}[Y|T=0, \boldsymbol{C}=f_C^\dagger(\boldsymbol{x})], \tag{2}$$

*with the Average Treatment Effect (ATE) of treatment $T$ defined as:*

$$ATE = \mathbb{E}[Y|do(T=1)] - \mathbb{E}[Y|do(T=0)] = \mathbb{E}_{p(\boldsymbol{C})}[\mathbb{E}[Y|T=1, \boldsymbol{C}] - \mathbb{E}[Y|T=0, \boldsymbol{C}]]. \tag{3}$$

Please note that we only consider the latent confounder component of the observed features $\boldsymbol{X}$ in the definition of CATE due to the indeterminate causal relation between the post-treatment variables $\boldsymbol{M}$ and the outcome $Y$. However, if the specific relationship between $\boldsymbol{M}$ and $Y$ can be further established by the researcher (e.g., all elements of $\boldsymbol{M}$ are latent mediators), more precise forms of CATE can be derived with path-specific counterfactual analysis (Cheng et al., 2022; Imai et al., 2010).

## 3.3 LATENT POST-TREATMENT BIAS

With $DEV(\boldsymbol{X}')$ (the ATE estimator that controls for the covariates $\boldsymbol{X}'$), CATE, and ATE defined in Section 3.2, in this section, we analyze the *latent post-treatment bias* of existing proxy-of-confounder-based causal inference methods, such as CEVAE, that control for latent variables inferred from the covariates $\boldsymbol{X}$ to estimate the ATE of $T$ on $Y$, when $\boldsymbol{X}$ scrambles both latent confounders and post-treatment variables as Assumption 1. In our analysis, Lemma 3.1 will be frequently used.

**Lemma 3.1.** *For an injective function $g$, $\mathbb{E}[Y|\boldsymbol{X}'=\boldsymbol{x}'] = \mathbb{E}[Y|g(\boldsymbol{X}')=g(\boldsymbol{x}')]$ holds.*

The proof when $g$ is differentiable *a.e.* can be referred to in Appendix A.1. Although the latent variable models used in existing methods (such as VAE with factorized Gaussian prior) lack identifiability guarantee (i.e., the recovery of the exact latent variables), we assume that these models can recover the true latent space $\boldsymbol{Z} = [\boldsymbol{C}||\boldsymbol{M}]$ up to invertible transformations $\bar{f}$, where the inference process can be represented as $\hat{\boldsymbol{Z}} = \tilde{f}(\boldsymbol{X}) = \bar{f} \circ f^\dagger(\boldsymbol{X})$. With such an assumption, we have the following theorem regarding the latent post-treatment bias when $\boldsymbol{X}$ mixes post-treatment variables.

**Theorem 3.2.** *If the observed covariates $\boldsymbol{X}$ are generated from latent confounders $\boldsymbol{C}$ and latent post-treatment variables $\boldsymbol{M}$ according to Assumption 1, the latent post-treatment bias of a proxy-based causal inference algorithm that controls latent variables $\hat{\boldsymbol{Z}}$ inferred from $\boldsymbol{X}$ via $\tilde{f} = \bar{f} \circ f^{\dagger} : \mathbb{R}^{K_X} \to \mathbb{R}^{K_C + K_M}$ to estimate the ATE can be formulated as follows:*

$$
\begin{aligned}
Bias(\boldsymbol{X}) &= ATE - DEV(\tilde{f}(\boldsymbol{X})) = ATE - \mathbb{E}[\mathbb{E}[Y|T=1, \tilde{f}(\boldsymbol{X})] - \mathbb{E}[Y|T=0, \tilde{f}(\boldsymbol{X})]] \\
&= ATE - \mathbb{E}[\mathbb{E}[Y|1, \bar{f} \circ f^{\dagger}(f(\boldsymbol{C}, \boldsymbol{M}))] - \mathbb{E}[Y|0, \bar{f} \circ f^{\dagger}(f(\boldsymbol{C}, \boldsymbol{M}))]] \\
&= \mathbb{E}[\mathbb{E}[Y|1, \boldsymbol{C}] - \mathbb{E}[Y|0, \boldsymbol{C}]] - \mathbb{E}[\mathbb{E}[Y|1, \boldsymbol{C}, \boldsymbol{M}] - \mathbb{E}[Y|0, \boldsymbol{C}, \boldsymbol{M}]],
\end{aligned}
\tag{4}
$$

*which can be arbitrarily bad. Therefore, the estimator of existing proxy-of-confounder-based methods, i.e., $DEV(\tilde{f}(\boldsymbol{X}))$, is an arbitrarily biased estimator of the ATE, when the selected proxy of confounders $\boldsymbol{X}$ accidentally mixes in latent post-treatment variables $\boldsymbol{M}$.*

The final step of Eq. (4) can be proved since $f$ is injective and $\bar{f}$ bijective, the composite $\bar{f} \circ f^{\dagger} \circ f : [\boldsymbol{C}||\boldsymbol{M}] \to \hat{\boldsymbol{Z}}$ is bijective, so we can use Lemma 3.1 to remove $\bar{f} \circ f^{\dagger} \circ f$ in the condition.

### 3.4 Examples in the Linear Case

Generally, the latent post-treatment bias defined in Eq. (4) cannot be simplified, because *(i)* the causal relationship between $\boldsymbol{M}$ and $Y$ are indeterminate, and *(ii)* the causal influence of $\boldsymbol{C}$, $\boldsymbol{M}$, and $T$ on $Y$ can be arbitrary. However, for linear structural causal models with determined causal relationships between $\boldsymbol{M}$ and $Y$ (e.g., $\boldsymbol{M}$ are mediators, which are post-treatment variables that have causal influences on the outcomes), stronger conclusions can be drawn as follows:

**Corollary 3.3.** (`Latent Mediator`). *For the linear Structural Causal Model (SCM) defined as:*

$$
\begin{aligned}
&(i)\ T \leftarrow \mathbb{1}(\alpha_T + \sum \beta_i \cdot C_i > a),\ (ii)\ M_j \leftarrow \alpha_M + \gamma_j \cdot T \\
&(iii)\ \boldsymbol{X} \leftarrow \boldsymbol{\alpha}_X + \mathbf{A}[\boldsymbol{C}||\boldsymbol{M}],\ (iv)\ Y \leftarrow \alpha_Y + \tau \cdot T + \sum \theta_j \cdot M_j + \sum \kappa_i \cdot C_i,
\end{aligned}
\tag{5}
$$

*where the mixture function $f = \mathbf{A} \in \mathbb{R}^{K_X \times (K_C + K_M)}$ is a full column-rank matrix, the CATE, ATE, and the bias of proxy-of-confounder-based causal inference model that controls the latent variables $\hat{\boldsymbol{Z}}$ inferred via $\hat{\boldsymbol{Z}} = \tilde{f}(\boldsymbol{X}) = \mathbf{B}^T \boldsymbol{X}$ can be formulated as follows:*

$$
\begin{aligned}
&ATE = CATE = \tau + \sum \gamma_j \cdot \theta_j, \text{ and } DEV(\hat{\boldsymbol{Z}}) = \mathbb{E}[DCEV(\hat{\boldsymbol{Z}})] = DCEV(\hat{\boldsymbol{Z}}) = \tau \\
&Bias(\hat{\boldsymbol{Z}}) = ATE - DEV(\hat{\boldsymbol{Z}}) = \sum \gamma_j \cdot \theta_j,
\end{aligned}
\tag{6}
$$

*where $\mathbf{B} \in \mathbb{R}^{K_X \times (K_C + K_M)}$ is another full column-rank matrix. Since $\sum \gamma_j \cdot \theta_j$ is arbitrary, the estimator $DEV(\hat{\boldsymbol{Z}}) = \mathbb{E}[DCEV(\mathbf{B}^T \boldsymbol{X})]$ is arbitrarily biased for ATE estimation.*

The proof of Eq. (6) is provided in Appendix A.2. In addition, we show that post-treatment variables $\boldsymbol{M}$ do NOT necessarily need to have direct causal effects on the outcome $Y$ to incur arbitrary bias in ATE estimation. In Appendix A.3, we provide another example (i.e., Latent Correlator) in the linear case where $\boldsymbol{M}$ is correlated with $Y$ through unobserved confounders $\boldsymbol{U}$ in Corollary A.1.

## 4 Methodology

In this section, we introduce the proposed Confounder-identifiable Variational Auto-Encoder (**CiVAE**) in detail. Specifically, we first prove that if the prior distribution of the true latent variables $\boldsymbol{Z} = [\boldsymbol{C}||\boldsymbol{M}]$ satisfies certain weak assumptions, CiVAE can *individually* identify $\boldsymbol{Z}$ up to element-wise bijective transformations and permutations. Then, utilizing the invariant causal relations between $\boldsymbol{C}$, $\boldsymbol{M}$, and $T$, we transform the challenging confounder disentanglement problem into a tractable pair-wise conditional independence test problem, which can be effectively solved with kernel-based methods, despite the potential arbitrary interactions among $\boldsymbol{C}$ and $\boldsymbol{M}$. Finally, we prove that controlling confounders inferred by CiVAE provides an unbiased estimate of ATE.

### 4.1 GENERATIVE PROCESS

We first introduce the generative process of $\boldsymbol{X}$ and $\boldsymbol{Z}$ in CiVAE that leads to the identification. The fundamental work on identifiability of deep variational inference, i.e., the identifiable VAE (iVAE) (Khemakhem et al., 2020), makes a strict assumption that the prior of true latent variables $\mathbf{Z}$ (i.e., $[C\|M]$) is conditionally factorized given observed covariates other than $\boldsymbol{X}$. However, since $\boldsymbol{C}$ form fork structures with $Y$, and $\boldsymbol{M}$ could have arbitrary relations with $Y$ (see Fig. 2-(c)), $C_i$, $C_j$, $M_i$, and $M_j$ are not independent given $Y$. Recently, Non-Factorized iVAE (NF-iVAE) (Lu et al., 2021) allows arbitrary dependence among the true latent variables $\boldsymbol{Z}$ in the conditional priors, where $\boldsymbol{Z}$ can be identified up to arbitrary non-linear transformations. However, the transformation is not necessarily invertible, which is risky as multiple values of confounders may collapse, leading to biased ATE estimation when averaging $DCEV$ calculated in each stratum of the inferred confounders.

In contrast, CiVAE guarantees the elementwise **bijective** identifiability of $\boldsymbol{Z}$ up to permutations. Specifically, we put a general exponential family *with at least one invertible sufficient statistic in the factorized part* as the prior when conditioning on treatment $T$ and outcome $Y$ as follows:

**Assumption 2.** *Let $\boldsymbol{Z} = [C\|M]$ be the random vector for latent variables that causally generate the observed covariates $\boldsymbol{X}$ according to Assumption 1. We assume that the conditional prior of $\boldsymbol{Z}$ given the outcome $Y$ and the treatment $T$ belongs to a general exponential family with parameter vector $\boldsymbol{\lambda}(Y, T)$ and sufficient statistics $\boldsymbol{S}(\boldsymbol{Z}) = [\boldsymbol{S}_f(\boldsymbol{Z})^T, \boldsymbol{S}_{nf}(\boldsymbol{Z})^T]^T$. Specifically, $\boldsymbol{S}(\boldsymbol{Z})$ is composed of (i) the sufficient statistics of a factorized exponential family, i.e., $\boldsymbol{S}_f(\boldsymbol{Z}) = [\boldsymbol{S}_{f,1}(Z_1)^T, \cdots, \boldsymbol{S}_{f,K_Z}(Z_{K_Z})^T]^T$, where all components $\boldsymbol{S}_{f,i}(Z_i)$ have dimension larger than or equal to 2 and **each $\boldsymbol{S}_{f,i}$ has at least one invertible dimension**, and (ii) $\boldsymbol{S}_{nf}(\boldsymbol{Z})$, where $\boldsymbol{S}_{nf}$ is a neural network with ReLU activation. The density of the conditional prior can be formulated as:*

$$p_{\boldsymbol{S},\boldsymbol{\lambda}}(\boldsymbol{Z}|Y, T) = \mathcal{Q}(\boldsymbol{Z})/\mathcal{C}(Y, T) \exp[\boldsymbol{S}(\boldsymbol{Z})^T \boldsymbol{\lambda}(Y, T)], \tag{7}$$

*where $\mathcal{Q}(\boldsymbol{Z})$ is the base measure, and $\mathcal{C}(Y, T)$ is the normalizing constant independent of $\boldsymbol{Z}$.*

**Intuition.** We justify that assumption 2 is practical as follows. *(i)* Neural networks with ReLU activation have **universal approximation ability** of distributions (Lu & Lu, 2020). Therefore, Eq. (7) can model arbitrary dependence between true latent confounders $\boldsymbol{C}$ and post-treatment variables $\boldsymbol{M}$ conditional on $T$ and $Y$. *(ii)* Although CiVAE makes an extra assumption that $\forall i$, at least one dimension of $\boldsymbol{S}_{f,i}$ is invertible, this can be easily satisfied as most commonly used exponential family distributions, such as Gaussian, Bernoulli, etc., have at least one invertible sufficient statistic.

The reason why we use ReLU as the activation is that, the identifiability of iVAE relies on the condition that the sufficient statistics $\boldsymbol{S}$ have zero second-order cross-derivative. The factorized part, i.e., $\boldsymbol{S}_f$, satisfies it trivially as all cross-derivatives of $\boldsymbol{S}_f$ are zero. In addition, since the ReLU neural networks are linear *a.e.*, all second-order derivatives of $\boldsymbol{S}_{nf}$ are zero. Therefore, identifiability holds after adding $\boldsymbol{S}_{nf}$ in the prior that allows the capturing of arbitrary dependence among $\boldsymbol{Z}$.

### 4.2 OPTIMIZATION OBJECTIVE

Combining Assumptions 1 and 2, the generative process assumed by CiVAE can be formulated as:

$$(i)\ p_{\boldsymbol{\theta}}(\boldsymbol{X}, \boldsymbol{Z} \mid Y, T) = p_f(\boldsymbol{X} \mid \boldsymbol{Z})\, p_{\boldsymbol{S},\boldsymbol{\lambda}}(\boldsymbol{Z} \mid Y, T),\ (ii)\ p_f(\boldsymbol{X} \mid \boldsymbol{Z}) = p_{\boldsymbol{\epsilon}}(\boldsymbol{X} - f(\boldsymbol{Z})). \tag{8}$$

where $\boldsymbol{\theta} = (f, \boldsymbol{\lambda}, \boldsymbol{S}) \in \Theta$ are the parameters of the generative distribution. $Y, T$ are omitted from $p_{\boldsymbol{\theta}}$ as $\boldsymbol{Z}$ form the Markov Blanket of $\boldsymbol{X}$ in Fig. 2-(c). Since the generative process of CiVAE is parameterized by deep neural networks, the posterior distribution of $\boldsymbol{Z}$, i.e., $p_{\boldsymbol{\theta}}(\boldsymbol{Z} \mid \boldsymbol{X}, Y, T)$, is intractable. Therefore, we resort to variational inference (Blei et al., 2017), where we introduce an approximate posterior $q_{\boldsymbol{\phi}}(\boldsymbol{Z} \mid \boldsymbol{X}, Y, T)$ parameterized by a deep neural network with trainable parameters $\boldsymbol{\phi}$, and in $q_{\boldsymbol{\phi}}(\boldsymbol{Z}|\cdot)$ finds the one closest to $p_{\boldsymbol{\theta}}(\boldsymbol{Z}|\cdot)$ measured by KL divergence. The minimization of KL is equivalent to the maximization of the evidence lower bound:

$$\mathcal{L}(\boldsymbol{\theta}, \boldsymbol{\phi}) := \mathbb{E}_{q_{\boldsymbol{\phi}}}\Big[ \log p_f(\boldsymbol{X} \mid \boldsymbol{Z}) + \underbrace{\log p_{\boldsymbol{S},\boldsymbol{\lambda}}(\boldsymbol{Z} \mid Y, T) - \log q_{\boldsymbol{\phi}}(\boldsymbol{Z} \mid \cdot)}_{\text{KL of posterior with prior}} \Big]. \tag{9}$$

Since the normalization constant $\mathcal{C}$ in Eq. (7) is generally intractable, it is infeasible to directly learn $\boldsymbol{S}, \boldsymbol{\lambda}$ by optimizing Eq. (9). Therefore, we substitute the KL term in Eq. (9) with the widely-used

score matching (Hyvärinen & Dayan, 2005) to learn unnormalized densities instead as follows:

$$\mathcal{L}(\boldsymbol{S}, \boldsymbol{\lambda}, \boldsymbol{\phi}) := \mathbb{E}_{q_\phi(\boldsymbol{Z}|\cdot)} \left[ \|\nabla_{\boldsymbol{Z}} \log q_\phi(\boldsymbol{Z} \mid \cdot) - \nabla_{\boldsymbol{Z}} \log p_{\boldsymbol{S}, \boldsymbol{\lambda}}(\boldsymbol{Z} \mid Y, T)\|^2 \right]$$

$$= \mathbb{E}_{q_\phi(\boldsymbol{Z}|\cdot)} \left[ \sum_{j=1}^{K_Z} \left[ \frac{\partial^2 p_{\boldsymbol{S}, \boldsymbol{\lambda}}(\boldsymbol{Z} \mid Y, T)}{\partial Z_j^2} + \frac{1}{2} \left( \frac{\partial p_{\boldsymbol{S}, \boldsymbol{\lambda}}(\boldsymbol{Z} \mid Y, T)}{\partial Z_j} \right)^2 \right] \right] + \text{const.} \tag{10}$$

## 4.3 IDENTIFIABILITY OF CIVAE

With the generative process and optimization objective of CiVAE discussed in previous sub-sections, we introduce the final assumption of CiVAE on the regularity of the data, which, combined with Assumptions 1 and 2, leads to the main Theorem that states the identifiability of CiVAE.

**Assumption 3.** *Assume the following: (i) The set $\{\boldsymbol{X} \in \mathcal{X} | \phi(\boldsymbol{X}) = 0\}$ has measure zero, where $\phi$ is the characteristic function of the density $p_f$ in Eq. (8). (ii) The sufficient statistics, $\boldsymbol{S}_{f,i}$ in $\boldsymbol{S}_f$ are all twice differentiable. (iii) The mixture function $f$ in Eq. (8) has all second-order cross derivatives. (iv) There exist $k + 1$ distinct points $(Y, T)_0, \cdots, (Y, T)_k$ s.t. the matrix $\mathbf{L} = [\boldsymbol{\lambda}((Y, T)_1) - \boldsymbol{\lambda}((Y, T)_0), \cdots, \boldsymbol{\lambda}((Y, T)_k) - \boldsymbol{\lambda}((Y, T)_0)]$ of size $k \times k$ is invertible, where $k = Dim(\boldsymbol{S})$.*

The identifiability theorem of CiVAE can be formulated as follows.

**Theorem 4.1.** *If Assumptions 1, 2, and 3 hold, and if $\boldsymbol{\theta}, \tilde{\boldsymbol{\theta}} \in \Theta \rightarrow p_{\boldsymbol{\theta}}(\boldsymbol{X}|Y, T) = p_{\tilde{\boldsymbol{\theta}}}(\boldsymbol{X}|Y, T)$, the true latent variables $\boldsymbol{Z}$ are identifiable up to **permutation** and **element-wise bijective transformation**. Furthermore, in the case of **variational inference**, if we denote the true parameter that generates the data as $\boldsymbol{\theta}^*$, if (i) the distribution family $q_\phi(\boldsymbol{Z}|\boldsymbol{X}, Y, T)$ contains the posterior $p_{\boldsymbol{\theta}}(\boldsymbol{Z}|\boldsymbol{X}, Y, T)$, and $q_\phi(\boldsymbol{Z}|\boldsymbol{X}, Y, T) > 0$, (ii) we optimize Eq. (4) w.r.t. both $\boldsymbol{\theta}, \phi$, then in the limit of infinite data, true parameters $\boldsymbol{\theta}^*$ can be learned up to a permutation and bijective transformation of $\boldsymbol{Z}$.*

**Intuition.** We justify that Assumption 3 is weak and practical as follows: 3-*(i)* is commonly-used to denote that the data generative distribution should not be degenerate. 3-*(ii)*, 3-*(iii)* can be trivially satisfied by neural networks. For 3-*(iv)*, Section B.2.3 of (Khemakhem et al., 2020) shows that if the functional for the factorized part of the exponential family parameters $\lambda_{ij}(Y, T)$ are independent (which is very weak), *(iv)* can be satisfied with arbitrary $k + 1$ different $(Y, T)$ points.

The proof of Theorem 4.1 extends NF-iVAE (Lu et al., 2021) by incorporating the new assumption introduced in CiVAE (i.e., each $\boldsymbol{S}_{f,i}$ has at least one invertible dimension) to ensure that the transformation of each $Z_i$ is bijective. The detailed proof is provided in Appendix A.4.

## 4.4 IDENTIFICATION OF LATENT CONFOUNDERS - NO INTERACTION CASE

Intuitively, theorem 4.1 ensures the latent variables $\hat{\boldsymbol{Z}}$ inferred by CiVAE cannot *(i)* mix confounders and post-treatment variables in each dimension, or *(ii)* collapse different values of the latent confounders into the same value. To further determine the dimensions of confounders and post-treatment variables in $\hat{\boldsymbol{Z}}$, we rely on the invariant causal relations between latent variables $\hat{\boldsymbol{Z}}$ and the treatment $T$. We first discuss the simplified cases with no latent interactions to gain intuitions:

- *Case 1. Intra-Confounders.* Latent confounders $C_i$, $C_j$ and the treatment $T$ form the *V-structure* $C_i \rightarrow T \leftarrow C_j$. Therefore, $C_i$ and $C_j$ are marginally **independent**, whereas they become **dependent** when conditioning on the assigned treatment $T$.

- *Case 2. Intra-Post Treatment Variables.* Latent post-treatment variables $M_i$, $M_j$ and the treatment $T$ form a *Fork-structure* $M_i \leftarrow T \rightarrow M_j$, where $M_i$, $M_j$ are marginally **dependent**, but they become **independent** after conditioning on the assigned treatment $T$.

- *Case 3. Cross-Confounder and Post-Treatment Variables.* Latent confounder $C_i$, latent post-treatment variable $M_j$, and the treatment $T$ forms a Chain structure $C_i \rightarrow T \rightarrow M_j$, where $C_i$, $M_j$ are marginally dependent, and they become **independent** after conditioning on $T$.

Since only in the case of *intra-confounders* does the dependence between two latent variables $\hat{Z}_i$ and $\hat{Z}_j$ **increase** after conditioning on the treatment $T$, if more than one latent confounder exists, which is highly probable when covariates $\boldsymbol{X}$ are high-dimensional, we can conduct independence

test $\mathrm{Ind}(\hat{Z}_i, \hat{Z}_j)$ and $\mathrm{CInd}(\hat{Z}_i, \hat{Z}_j|T)$ for all pairs of inferred latent variables in $\hat{\boldsymbol{Z}}$ and select the pairs where the p-value of $\mathrm{CInd}$ is larger than that of $\mathrm{Ind}$ as latent confounders. Here, we note that the kernel-based (conditional) independence test incurs $N^2 \times K_Z^2$ complexity in the training phase (Zhang et al., 2012). However, once the dimensions of the confounders in $\hat{\boldsymbol{Z}}$ are determined, CiVAE **has the same complexity as CEVAE** for the estimation of CATE and ATE in the test phase.

### 4.5 GENERALIZATION TO INTERACTED LATENT VARIABLES

We further generalize CiVAE to address **interactions among latent variables** $[\boldsymbol{C}\|\boldsymbol{M}]$. Since Assumption 2 allows arbitrary dependence in the latent space, CiVAE can still individually identify $[\boldsymbol{C}\|\boldsymbol{M}]$ in $\hat{\boldsymbol{Z}}$ by optimizing Eq. (10). However, in scenarios where latent interactions exist, *cases 1-3* in Section 4.4 may not hold, which precludes us from further disentangling $\boldsymbol{C}$ from $\hat{\boldsymbol{Z}}$. For example, if $\boldsymbol{C}$ confounds $\boldsymbol{M}$, *case 2* may not hold, as $M_i$ and $M_j$ are still dependent after conditioning on $T$.

The generalization still leverages the *V-structure* between latent confounders $\boldsymbol{C}$ and treatment $T$, i.e., $C_i \to T \leftarrow C_j$ and involves more advanced pairwise independence test for every $\hat{Z}_i, \hat{Z}_j \in \hat{\boldsymbol{Z}}$:

- ***Case 1\****. *Intra-Confounders.* If there exist $\hat{\boldsymbol{Z}}_C \subset \hat{\boldsymbol{Z}}/\{\hat{Z}_i, \hat{Z}_j\}, s.t.\ \hat{Z}_i \perp \hat{Z}_j|\hat{\boldsymbol{Z}}_C$.

- ***Case 2\****. *Intra-Post Treat. Variables.* If there exist $\hat{\boldsymbol{Z}}_M \subset \hat{\boldsymbol{Z}}/\{\hat{Z}_i, \hat{Z}_j\}, s.t.\ \hat{Z}_i \perp \hat{Z}_j|\{\hat{\boldsymbol{Z}}_M, T\}$.

- ***Case 3\****. *Cross-Variables.* If there exist $\hat{\boldsymbol{Z}}_{C,M} \subset \hat{\boldsymbol{Z}}/\{\hat{Z}_i, \hat{Z}_j\}, s.t.\ \hat{Z}_i \perp \hat{Z}_j|\{\hat{\boldsymbol{Z}}_{C,M}, T\}$.

The proof is that, for ***Case 2\**** and ***Case 3\****, since $M_i, M_j$ form fork structure with $T$ and $M_i, C_j$ form chain structure with $T$, we cannot find $\hat{\boldsymbol{Z}}_C$ that leads to independence without conditioning on $T$ (as with ***Case 1\****). If no latent interactions exist, $\hat{\boldsymbol{Z}}_C, \hat{\boldsymbol{Z}}_M, \hat{\boldsymbol{Z}}_{C,M} = \emptyset$, which degenerates to the three cases discussed in Section 4.4. If interaction exists, we start with $\hat{\boldsymbol{Z}}_C, \hat{\boldsymbol{Z}}_M, \hat{\boldsymbol{Z}}_{C,M} = \emptyset$ and gradually include more variables in $\hat{\boldsymbol{Z}}$ into the condition sets. We provide examples of $\hat{\boldsymbol{Z}}_C, \hat{\boldsymbol{Z}}_M, \hat{\boldsymbol{Z}}_{C,M}$ for different types of latent interactions in Appendix C. Similarly, once the dimensions of the confounders $\hat{\boldsymbol{C}}$ in $\hat{\boldsymbol{Z}}$ are determined, retrieving $\hat{\boldsymbol{C}}$ from $\hat{\boldsymbol{Z}}$ takes $O(1)$ time in the test phase.

### 4.6 ATE ESTIMATOR WITH TRANSFORMED CONFOUNDERS

Finally, we demonstrate that controlling the transformed confounders $\hat{\boldsymbol{C}}$ inferred by CiVAE provides an unbiased estimation of ATE. Specifically, we have the final Theorem show the unbiasedness.

**Theorem 4.2.** *Controlling bijective of confounders is equivalent to original confounders in ATE estimation, i.e., $DEV(\tilde{\boldsymbol{C}}) = DEV(g(\boldsymbol{C})) = ATE$, if the transformation function $g$ is bijective.*

The proof of Theorem 4.2 for discrete $\boldsymbol{C}$ is trivial (where $\hat{\boldsymbol{C}} = g(\boldsymbol{C})$ represents a simple relabeling of the stratum that we calculate the $DCEV$ and take the expectation). With Theorem 4.2, we can control the identified latent confounders as true confounders, providing an unbiased estimate of ATE.

## 5 EMPIRICAL STUDY

### 5.1 DATASETS

**Simulated Datasets.** We first establish two simulated datasets, i.e., LatentMediator and LatentCorrelator, that consider two types of post-treatment variables, i.e., *(i)* mediators and *(ii)* correlators, i.e., variables that are correlated with the outcome $Y$ via latent confounders $\boldsymbol{U}$. The generative processes of the two datasets are provided in Corollary 3.3 and Corollary A.1. In our experiments, $\boldsymbol{C}$ are generated from Gaussian distribution as $\boldsymbol{C} \sim Gaussian(0, \mathbf{I}_{K_C})$. For LatentMediator, $\boldsymbol{\gamma}$ is set as $[-1, -1, -1]$, $\boldsymbol{\theta}$ is set as $[1, 1, 1]$, and $\tau$ is set as 2, which results in an $ATE = -1$. For the

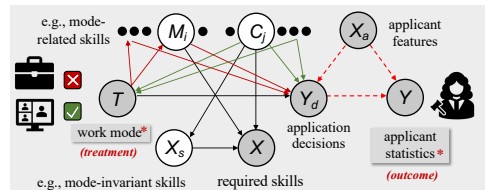

Figure 3: Generative process of the real-world datasets.

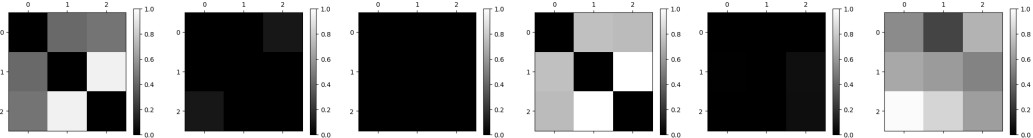

(a) **Case 1:** Intra-Confounder    (b) **Case 2:** Intra-Mediator    (c) **Case 3:** Confounder-Mediator

Figure 4: Visualization of $p$-value of independence test before and after conditioning on treatment $T$.

`LatentCorrelator` dataset, we set the same $\gamma$ and $\theta$ as the `LatentMediator` dataset, where parameters $\phi$ and $\tau$ are set to 1, which results in $ATE$ of 1.

**Real-world Datasets.** In addition, we build real-world datasets from LinkedIn to estimate the ATE of *switching a job from **onsite** to **online** work mode* on *the decision of applicants*. The decision can be measured by the click-through rate of applicants to the job posting (i.e., $Y_d$ in Fig. 3), which is too coarse in granularity. More practically, we consider the average age and the variance of gender of the applicants as two outcomes of interest, which are determined by the direct outcome, i.e., decision to apply $Y_d$, and applicants' age and gender $X_a$, where $X_a$ are pre-treatment variables. The causal graph is illustrated in Fig. 3, which extends the example in Fig. 1. Covariates $\boldsymbol{X} \in \{0,1\}^{K_X}$ include the required skills of the job. Specifically, we establish a cohort of 3,228 jobs from the Bay Area in the US, where a preliminary study shows that $DEV(\emptyset) \approx 2$ years[2] (i.e., online job applicants are two years younger than onsite job applicants in the collected data), and $DEV(\emptyset) \approx -0.015$ (i.e., online jobs exhibit 0.015 more gender variance than onsite jobs in the collected data). To simulate $\boldsymbol{C}$ and $\boldsymbol{M}$, we first learn a generative model as follows:

$$\boldsymbol{Z} \sim Gaussian(\boldsymbol{0}, \mathbf{I}_{K_Z}), \boldsymbol{X} \sim Multi(NN_f(\boldsymbol{Z})), Y \sim Gaussian(\boldsymbol{w} \odot \boldsymbol{Z}, 1), \quad (11)$$

where $Multi$ represents multinomial distribution, $NN_f$ is a neural network with softmax activation, $\boldsymbol{Z}, \boldsymbol{w} \in \mathbb{R}^{K_Z}$, $K_Z = 8$, and $\odot$ represents the element-wise product operator, respectively. We then treat the first $K_C = 5$ dimensions of $\boldsymbol{Z}$ as the latent confounders $\boldsymbol{C}$ and the remaining $K_M = K_Z - K_C$ dimensions as the latent mediators $\boldsymbol{M}$. After learning $NN_f$ and $\boldsymbol{w}$ according to Eq. (11), we draw latent confounders $\boldsymbol{C} \in Gaussian(\boldsymbol{0}, \mathbf{I})$, latent mediators $\boldsymbol{M} = T \cdot \gamma$, and set the outcome $Y = \boldsymbol{w} \odot [\boldsymbol{C} || \boldsymbol{M}] + \tau \cdot T$, where the true ATE can be calculated as $sum(\gamma \odot \boldsymbol{w}_{-K_M:}) + \tau$.

### 5.1.1 DISENTANGLE CONFOUNDERS AND POST-TREATMENT VARIABLES

We first show the $p$-value of the kernel-based pairwise independence test of the true latent variables before and after conditioning on the assigned treatment $T$. From Fig. 4, we can find that the distinction of the intra-confounder case from the other two cases discussed in Subsection 4.4 is significant. Here, we should note this relies on the assumption that latent variables are independent. Experiments on generalized CiVAE to address interactions among latent variables are in Section C.

### 5.2 BASELINES

The baselines we include can be categorized into three classes. *(i)* **Unawareness**, where no information in $\boldsymbol{X}$ is used for ATE estimation. We implement the naive LR0 estimator, which regresses $Y$ on $T$ and uses the coefficient to estimate the ATE (Imbens & Rubin, 2015) and is equivalent to $DEV(\emptyset)$. *(ii)* **Control-$\boldsymbol{X}$**, which directly controls the covariates $\boldsymbol{X}$. In this class, LR1 regresses $Y$ on $T$ and $\boldsymbol{X}$, whereas TarNet uses a two-branch neural network to estimate $DEV(\boldsymbol{X})$ *(iii)* **Control-$\boldsymbol{Z}$**, which controls latent variables $\boldsymbol{Z}$ learned from the covariates $\boldsymbol{X}$. Methods from this class include CEVAE (Louizos et al., 2017) and covariate disentanglement methods, such as DR-CFR (Hassanpour & Greiner, 2020), TEDVAE (Zhang et al., 2021), NICE (Shi et al., 2021), and AFS (Wang et al., 2023).

### 5.2.1 RESULTS AND ANALYSIS

From Table 1, we can find that for all four datasets, CEVAE is worse than the naive LR0 estimator. In addition, for the `LatentMediator` and `LinkedIn(Age)` datasets, all methods except CiVAE fail to predict the negativity of the ATE. Covariates disentanglement-based methods, i.e., DR-CFR and TEDVAE, inherit the latent post-treatment bias of CEVAE. The reason is that, these methods

---

[2]which leads to 0.178 and -0.105 after standardization of the outcome.

Table 1: Comparison of CiVAE with baselines under latent post-treatment bias on various datasets.

| Dataset | LatentMediator | | LatentCorrelator | | LinkedIn(Age) | | LinkedIn(Gender) | |
|---|---|---|---|---|---|---|---|---|
| Method | ATE. | Err. | ATE. | Err. | ATE. | Err. | ATE. | Err. |
| LR0 | $0.975 \pm 0.032$ | 1.975 | $2.977 \pm 0.032$ | 1.977 | $0.131 \pm 0.015$ | 0.399 | $-0.105 \pm 0.009$ | -0.213 |
| LR1 | $1.457 \pm 0.167$ | 2.457 | $3.400 \pm 0.130$ | 2.400 | $0.093 \pm 0.029$ | 0.361 | $-0.175 \pm 0.014$ | -0.256 |
| TarNet | $1.461 \pm 0.172$ | 2.461 | $3.414 \pm 0.146$ | 2.414 | $0.112 \pm 0.085$ | 0.380 | $-0.167 \pm 0.021$ | -0.248 |
| CEVAE | $1.550 \pm 0.292$ | 2.550 | $3.323 \pm 0.167$ | 2.323 | $0.106 \pm 0.078$ | 0.374 | $-0.180 \pm 0.028$ | -0.261 |
| DR-CFR | $1.239 \pm 0.324$ | 2.239 | $3.185 \pm 0.319$ | 2.185 | $0.094 \pm 0.089$ | 0.362 | $-0.159 \pm 0.030$ | -0.240 |
| NICE | $1.868 \pm 0.530$ | 2.868 | $1.942 \pm 0.524$ | 0.942 | $0.149 \pm 0.126$ | 0.417 | $-0.186 \pm 0.041$ | -0.267 |
| TEDVAE | $1.042 \pm 0.315$ | 2.042 | $3.138 \pm 0.281$ | 2.138 | $0.097 \pm 0.093$ | 0.365 | $-0.143 \pm 0.027$ | -0.224 |
| AFS | $1.496 \pm 0.825$ | 2.496 | $3.251 \pm 0.398$ | 2.251 | $0.105 \pm 0.102$ | 0.373 | $-0.163 \pm 0.045$ | -0.244 |
| CiVAE | $\mathbf{-0.822} \pm 0.753$ | **0.178** | $\mathbf{1.199} \pm 0.765$ | **0.199** | $\mathbf{-0.140} \pm 0.137$ | **0.128** | $\mathbf{-0.106} \pm 0.064$ | **-0.187** |
| True ATE | $-1.000 \pm 0.000$ | 0.000 | $1.000 \pm 0.000$ | 0.000 | $-0.268 \pm 0.000$ | 0.000 | $-0.081 \pm 0.000$ | 0.000 |

disentangle latent confounders $C$ from latent instrumental variables $I$ and latent adjusters $A$ by utilizing their causal relations with $T$ and $Y$, i.e., $I$ is predictive only for $T$, $A$ is predictive only for $Y$, whereas $C$ is predictive for both $T$ and $Y$. For example, TEDVAE includes three encoders to infer three sets of latent variables $\hat{I}$, $\hat{A}$, $\hat{C}$ from $X$ and adds classification losses $p(T|\hat{I},\hat{C})$ and $p(Y|T,\hat{C},\hat{A})$ on the CEVAE loss. However, since both $C$ and $M$ are correlated with both $T$ and $Y$, these methods cannot disentangle $C$ from $M$. An exception is NICE (Shi et al., 2021), which uses invariant risk minimization (IRM) (Arjovsky et al., 2019) to find all causal parents of the outcome $Y$ as the confounders, which makes it more robust in the `LatentCorrelator` case. However, since mediators $M$ are also the causal parent of $Y$, the performance degrades substantially on the `LatentMediator` dataset. Although AFS (Wang et al., 2023) considers the existence of post-treatment variables $M$ in the proxy $X$, it assumes that they can be separated from other variables in $X$ in the observational space, and no relationship exists between the post-treatment variables and the outcome, so it still has poor performance in our setting since both assumptions are violated.

## 5.3 SENSITIVITY ANALYSIS

In this part, we vary the number of confounders and post-treatment variables that generate proxy $X$ in the `LinkedIn(Age)` and `LinkedIn(Gender)` datasets and compare CiVAE with the baseline TEDVAE in Fig. 5. Fig. 5 shows that the error is consistently lower for CiVAE. In addition, the error is comparatively higher when the number of confounders is low since the mis-identification of latent post-treatment variables as confounders

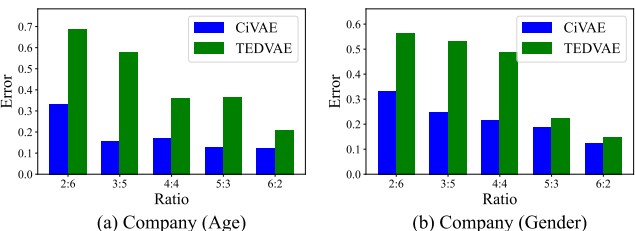

(a) Company (Age)  (b) Company (Gender)

Figure 5: Error with different ratio of latent confounders and latent post-treatment variable in the latent space.

can have a comparatively larger influence on the ATE estimation. In addition, when the number of confounders becomes larger, the performance gap between CiVAE and TEDVAE gracefully shrinks.

## 6 CONCLUSIONS

In this paper, we systematically investigate the latent post-treatment bias in causal inference from observational data. We first prove that unresolved latent post-treatment variables scrambled in the proxy of confounders can arbitrarily bias the ATE estimation. To address the bias, we proposed the Confounder-identifiable VAE (CiVAE), which, utilizing a mild assumption regarding the prior of latent factors, guarantees the identifiability of latent confounders up to bijective transformations. Finally, we show that controlling the latent confounders inferred by CiVAE can provide an unbiased estimation of the ATE. Experiments on both simulated and real-world datasets demonstrate that CiVAE has superior robustness to latent post-treatment bias compared to state-of-the-art methods.

ACKNOWLEDGMENT

This work is supported in part by the National Science Foundation (NSF) under grants IIS-2006844, IIS-2144209, IIS-2223769, CNS-2154962, BCS-2228534, and CMMI-2411248; the Office of Naval Research (ONR) under grant N000142412636; and the Commonwealth Cyber Initiative (CCI) under grant VV-1Q24-011.

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

# Appendix

## A THEORETICAL ANALYSIS

### A.1 PROOF OF LEMMA 3.1.

*Proof.* Let $\boldsymbol{Z} = g(\boldsymbol{X})$ and $\boldsymbol{z} = g(\boldsymbol{x})$. If $g$ is injective and differentiable *a.e.*, and $g^\dagger$ is the left-inverse, we have:

$$f_{Y|g(\boldsymbol{X})}(y|g(\boldsymbol{x})) = f_{Y|\boldsymbol{Z}}(y|\boldsymbol{z}) = \frac{f_{Y,\boldsymbol{Z}}(y,\boldsymbol{z})}{f_{\boldsymbol{Z}}(\boldsymbol{z})} = \frac{f_{Y,\boldsymbol{X}}(y,g^\dagger(\boldsymbol{z}))|\mathbf{J}_{g^\dagger}(\boldsymbol{z})|}{f_{\boldsymbol{X}}(g^\dagger(\boldsymbol{z}))|\mathbf{J}_{g^\dagger}(\boldsymbol{z})|} = \frac{f_{Y,\boldsymbol{X}}(y,\boldsymbol{x})}{f_{\boldsymbol{X}}(\boldsymbol{x})} = f_{Y|\boldsymbol{X}}(y|\boldsymbol{x}),$$
(12)

where $f_{\cdot}$ and $f_{\cdot|\cdot}$ represent the marginal and conditional density functions, respectively, and $\mathbf{J}_{g^\dagger}(\boldsymbol{z})$ is the Jacobian matrix of function $g^\dagger$ evaluated at $\boldsymbol{z}$. Based on Eq. (12), we have:

$$\mathbb{E}[Y|\boldsymbol{X}] = \int \boldsymbol{y} \cdot f_{Y|\boldsymbol{X}}(\boldsymbol{y}|\boldsymbol{x}) dy = \int y \cdot f_{Y|\boldsymbol{Z}}(\boldsymbol{y}|\boldsymbol{z}) dy = \mathbb{E}[Y|\boldsymbol{Z} = \boldsymbol{z}] = \mathbb{E}[Y|g(\boldsymbol{X}) = g(\boldsymbol{x})]. \quad (13)$$

$\square$

### A.2 PROOF OF COROLLARY 3.3.

*Proof.* For $\boldsymbol{X} = \boldsymbol{x}$, let $[\boldsymbol{c}||\boldsymbol{m}] \doteq [f_C^\dagger(\boldsymbol{x})||f_M^\dagger(\boldsymbol{x})] \doteq f^\dagger(\boldsymbol{x}) = \mathbf{A}^\dagger(\boldsymbol{x} - \boldsymbol{\alpha}_X)$, where $\mathbf{A}^\dagger$ is the left inverse of the full column-rank matrix $\mathbf{A}$ in Eq. (2), we have:

$$
\begin{aligned}
CATE(\boldsymbol{x}) &= \mathbb{E}[Y|T = 1, \boldsymbol{C} = f_C^\dagger(\boldsymbol{x})] - \mathbb{E}[Y|T = 0, \boldsymbol{C} = f_C^\dagger(\boldsymbol{x})] \\
&= \mathbb{E}[Y|T = 1, \boldsymbol{C} = \boldsymbol{c}] - \mathbb{E}[Y|T = 0, \boldsymbol{C} = \boldsymbol{c}] \\
&= \mathbb{E}[\alpha_Y + \tau \cdot T + \sum \theta_j \cdot M_j + \sum \kappa_i \cdot C_i | T = 1, \boldsymbol{C} = \boldsymbol{c}] \\
&\quad - \mathbb{E}[\alpha_Y + \tau \cdot T + \sum \theta_j \cdot M_j + \sum \kappa_i \cdot C_i | T = 0, \boldsymbol{C} = \boldsymbol{c}] \\
&= \alpha_Y + \tau \cdot \mathbb{E}[T|T = 1, \boldsymbol{C} = \boldsymbol{c}] + \sum \theta_j \cdot \mathbb{E}[M_j|T = 1, \boldsymbol{C} = \boldsymbol{c}] + \sum \kappa_i \cdot \mathbb{E}[C_i|T = 1, \boldsymbol{C} = \boldsymbol{c}] \\
&\quad - \alpha_Y + \tau \cdot \mathbb{E}[T|T = 0, \boldsymbol{C} = \boldsymbol{c}] + \sum \theta_j \cdot \mathbb{E}[M_j|T = 0, \boldsymbol{C} = \boldsymbol{c}] + \sum \kappa_i \cdot \mathbb{E}[C_i|T = 0, \boldsymbol{C} = \boldsymbol{c}] \\
&= \tau \cdot (1 - 0) + \sum \theta_j \cdot (\gamma_j \cdot (1 - 0)) + \sum \kappa_i \cdot (c_i - c_i) \\
&= \tau + \sum \theta_j \cdot \gamma_j = \mathbb{E}[\tau + \sum \theta_j \cdot \gamma_j] = ATE,
\end{aligned}
$$
(14)

where the first equality is due to the definition of CATE in Eq. (2). In addition, the causal estimand and bias of a proxy-of-confounder-based causal inference model that controls the latent variable $\boldsymbol{Z}$ inferred via $\boldsymbol{Z} = \bar{f}(\boldsymbol{X}) = \mathbf{B}^T \boldsymbol{X}$ (where $\mathbf{B}$ is also a full column-rank matrix) can be formulated as:

$$
\begin{aligned}
DCEV(\mathbf{B}^T \boldsymbol{x}) &= \mathbb{E}[Y|T = 1, \boldsymbol{Z} = \mathbf{B}^T \boldsymbol{x}] - \mathbb{E}[Y|T = 0, \boldsymbol{Z} = \mathbf{B}^T \boldsymbol{x}] \\
&= \mathbb{E}[Y|T = 1, \boldsymbol{Z} = \mathbf{B}^T \boldsymbol{\alpha}_X + \mathbf{B}^T \mathbf{A}[\boldsymbol{c}||\boldsymbol{m}]] - \mathbb{E}[Y|T = 0, \boldsymbol{Z} = \mathbf{B}^T \boldsymbol{\alpha}_X + \mathbf{B}^T \mathbf{A}[\boldsymbol{c}||\boldsymbol{m}]] \\
&\stackrel{(a)}{=} \mathbb{E}[Y|T = 1, \boldsymbol{C} = \boldsymbol{c}, \boldsymbol{M} = \boldsymbol{m}] - \mathbb{E}[Y|T = 0, \boldsymbol{C} = \boldsymbol{c}, \boldsymbol{M} = \boldsymbol{m}] \\
&= \alpha_Y + \tau \cdot 1 + \sum \theta_j \cdot \mathbb{E}[M_j|T = 1, \boldsymbol{C} = \boldsymbol{c}, \boldsymbol{M} = \boldsymbol{m}] + \sum \kappa_i \cdot \mathbb{E}[C_i|T = 1, \boldsymbol{C} = \boldsymbol{c}, \boldsymbol{M} = \boldsymbol{m}] \\
&\quad - \alpha_Y + \tau \cdot 0 + \sum \theta_j \cdot \mathbb{E}[M_j|T = 0, \boldsymbol{C} = \boldsymbol{c}, \boldsymbol{M} = \boldsymbol{m}] + \sum \kappa_i \cdot \mathbb{E}[C_i|T = 0, \boldsymbol{C} = \boldsymbol{c}, \boldsymbol{M} = \boldsymbol{m}] \\
&= \tau \cdot (1 - 0) + \sum \theta_j \cdot (m_j - m_j) + \sum \kappa_i \cdot (c_i - c_i) \\
&= \tau = \mathbb{E}[\tau] = \mathbb{E}[DCEV(\mathbf{B}^T \boldsymbol{X})],
\end{aligned}
$$
(15)

where step (a) is due to the fact that, since both $\mathbf{A}$ and $\mathbf{B}$ are full column-rank matrices, $\mathbf{B}^T \mathbf{A}$ is an invertible matrix, and the mapping $f = \mathbf{B}^T \boldsymbol{\alpha}_X + \mathbf{B}^T \mathbf{A}$ is bijective. Therefore, we can invoke Lemma 3.1 and apply the left-inverse of $f$, i.e., $f^\dagger = (\mathbf{B}^T \mathbf{A})^{-1} - \mathbf{B}^T \boldsymbol{\alpha}_X$, to the condition of the expectation. The rest steps are based on the structural causal equations defined in Eq. (2). $\square$

## A.3 Another Case of Linear SCM with Latent Correlators

**Corollary A.1.** *(`Latent Correlator`). For another Linear Structural Causal Model defined as:*

$$
\begin{aligned}
T &\leftarrow \mathbb{1}(\alpha_T + \sum \beta_i \cdot C_i > a) \\
M_j &\leftarrow \alpha_M + \gamma_j \cdot T + \phi_j \cdot U_j \\
\boldsymbol{X} &\leftarrow \boldsymbol{\alpha}_X + \mathbf{A}[\boldsymbol{M}||\boldsymbol{C}] \\
Y &\leftarrow \alpha_Y + \tau \cdot T + \sum \theta_j \cdot U_j + \sum \kappa_i \cdot C_i,
\end{aligned}
\tag{16}
$$

*where $f = \mathbf{A} \in \mathbb{R}^{K_X \times (K_C + K_M)}$ is a full column-rank matrix, the CATE, ATE, and the bias of proxy-of-confounder-based causal inference model that controls the latent variable $\boldsymbol{Z}$ inferred via $\boldsymbol{Z} = \bar{f}(\boldsymbol{X}) = \mathbf{B}^T \boldsymbol{X}$ can be formulated as follows:*

$$
\begin{aligned}
ATE &= CATE = \tau \\
\mathbb{E}[DCEV(\boldsymbol{Z} = \mathbf{B}^T \boldsymbol{X})] &= DCEV(\boldsymbol{Z} = \mathbf{B}^T \boldsymbol{X}) = \tau - \sum \frac{\theta_j \cdot \gamma_j}{\phi_j} \\
Bias &= ATE - \mathbb{E}[DCEV(\mathbf{B}^T \boldsymbol{X})] = \sum \frac{\theta_j \cdot \gamma_j}{\phi_j},
\end{aligned}
\tag{17}
$$

*where $\mathbf{B} \in \mathbb{R}^{K_X \times (K_C + K_M)}$ is another full column-rank matrix. Since $\sum \frac{\theta_j \cdot \gamma_j}{\phi_j}$ is arbitrary, the estimator $\mathbb{E}[DCEV(\mathbf{B}^T \boldsymbol{X})]$ is arbitrarily biased for the estimation of ATE.*

*Proof.* The proof of the CATE and ATE is trivial. The causal estimand and the bias of a proxy-of-confounder-based causal inference model that controls the latent variables $\boldsymbol{Z}$ inferred via $\boldsymbol{Z} = \bar{f}(\boldsymbol{X}) = \mathbf{B}^T \boldsymbol{X}$ (where $\mathbf{B}$ is also a full column-rank matrix) can be formulated as follows:

$$
\begin{aligned}
DCEV(\mathbf{B}^T \boldsymbol{x}) &= \mathbb{E}[Y|T=1, \boldsymbol{Z} = \mathbf{B}^T \boldsymbol{x}] - \mathbb{E}[Y|T=0, \boldsymbol{Z} = \mathbf{B}^T \boldsymbol{x}] \\
&= \mathbb{E}[Y|T=1, \boldsymbol{Z} = \boldsymbol{\alpha}_X + \mathbf{B}^T \mathbf{A}[\boldsymbol{c}||\boldsymbol{m}]] - \mathbb{E}[Y|T=0, \boldsymbol{Z} = \boldsymbol{\alpha}_X + \mathbf{B}^T \mathbf{A}[\boldsymbol{c}||\boldsymbol{m}]] \\
&\overset{(a)}{=} \mathbb{E}[Y|T=1, \boldsymbol{C} = \boldsymbol{c}, \boldsymbol{M} = \boldsymbol{m}] - \mathbb{E}[Y|T=0, \boldsymbol{C} = \boldsymbol{c}, \boldsymbol{M} = \boldsymbol{m}] \\
&= \alpha_Y + \tau \cdot 1 + \sum \theta_j \cdot \mathbb{E}[U_j|T=1, \boldsymbol{C} = \boldsymbol{c}, \boldsymbol{M} = \boldsymbol{m}] + \sum \kappa_i \cdot \mathbb{E}[C_i|T=1, \boldsymbol{C} = \boldsymbol{c}, \boldsymbol{M} = \boldsymbol{m}] \\
&\quad - \alpha_Y + \tau \cdot 0 + \sum \theta_j \cdot \mathbb{E}[U_j|T=0, \boldsymbol{C} = \boldsymbol{c}, \boldsymbol{M} = \boldsymbol{m}] + \sum \kappa_i \cdot \mathbb{E}[C_i|T=0, \boldsymbol{C} = \boldsymbol{c}, \boldsymbol{M} = \boldsymbol{m}] \\
&= \tau \cdot (1-0) + \sum \theta_j \cdot (\phi_j^{-1} \cdot (m_j - \alpha_M - \gamma_j) - \phi_j^{-1} \cdot (m_j - \alpha_M)) + \sum \kappa_i \cdot (c_i - c_i) \\
&= \tau - \sum \frac{\theta_j \cdot \gamma_j}{\phi_j} = \mathbb{E}\left[\tau - \sum \frac{\theta_j \cdot \gamma_j}{\phi_j}\right] = \mathbb{E}[DCEV(\mathbf{B}^T \boldsymbol{X})],
\end{aligned}
\tag{18}
$$

$\square$

where step (a) and the rest of the proof follow the same logic as the proof in Section 3.3.

## A.4 Proof of Theorem 4.1

The strict definitions of the exponential family, strong exponential (which is assumed for the factorized part of the conditional prior), and identifiability follow the definitions in (Khemakhem et al., 2020; Lu et al., 2021), and can be referred to in Appendix E, F of (Lu et al., 2021), which we omit to avoid redundancy. The proof of Theorem 4.1 is largely based on the NF-iVAE paper (Lu et al., 2021), where most of the details can be found, with the new assumption introduced in CiVAE that each $\boldsymbol{S}_{f,i}$ has at least one invertible dimension incorporated to ensure that each dimension of the inferred latent variables is a bijective transformation of the corresponding true latent variable.

### A.4.1 PART I

**Step I**. In this step, we transform the equality of noisy conditional marginal distribution of $\boldsymbol{X}$ given $Y, T$ of two models with parameter $\boldsymbol{\theta}, \tilde{\boldsymbol{\theta}} \in \Theta$ into the equality of noise-free distributions.

$$
\begin{aligned}
& p_{\boldsymbol{\theta}}(\boldsymbol{X} \mid Y, T) = p_{\tilde{\boldsymbol{\theta}}}(\boldsymbol{X} \mid Y, T) \\
\Longrightarrow & \int_{\mathcal{Z}} p_f(\boldsymbol{X} \mid \boldsymbol{Z}) p_{\boldsymbol{S}, \boldsymbol{\lambda}}(\boldsymbol{Z} \mid Y, T) d\boldsymbol{Z} = \int_{\mathcal{Z}} p_{\tilde{f}}(\boldsymbol{X} \mid \boldsymbol{Z}) p_{\tilde{\boldsymbol{S}}, \tilde{\boldsymbol{\lambda}}}(\boldsymbol{Z} \mid Y, T) d\boldsymbol{Z} \\
\Longrightarrow & \int_{\mathcal{Z}} p_{\boldsymbol{\varepsilon}}(\boldsymbol{X} - f(\boldsymbol{Z})) p_{\boldsymbol{S}, \boldsymbol{\lambda}}(\boldsymbol{Z} \mid Y, T) d\boldsymbol{Z} = \int_{\mathcal{Z}} p_{\boldsymbol{\varepsilon}}(\boldsymbol{X} - \tilde{f}(\boldsymbol{Z})) p_{\tilde{\boldsymbol{S}}, \tilde{\boldsymbol{\lambda}}}(\boldsymbol{Z} \mid Y, T) d\boldsymbol{Z} \\
\stackrel{(a)}{\Longrightarrow} & \int_{\mathcal{X}} p_{\boldsymbol{\varepsilon}}(\boldsymbol{X} - \overline{\boldsymbol{X}}) p_{\boldsymbol{S}, \boldsymbol{\lambda}}\left(f^{\dagger}(\overline{\boldsymbol{X}}) \mid Y, T\right) \operatorname{vol}\left(\mathbf{J}_{f^{\dagger}}(\overline{\boldsymbol{X}})\right) d\overline{\boldsymbol{X}} = \\
& \int_{\mathcal{X}} p_{\boldsymbol{\varepsilon}}(\boldsymbol{X} - \overline{\boldsymbol{X}}) p_{\tilde{\boldsymbol{S}}, \tilde{\boldsymbol{\lambda}}}\left(\tilde{f}^{\dagger}(\overline{\boldsymbol{X}}) \mid Y, T\right) \operatorname{vol}\left(\mathbf{J}_{\tilde{f}^{\dagger}}(\overline{\boldsymbol{X}})\right) d\overline{\boldsymbol{X}} \\
\stackrel{(b)}{\Longrightarrow} & \int_{\mathbb{R}^d} p_{\boldsymbol{\varepsilon}}(\boldsymbol{X} - \overline{\boldsymbol{X}}) \tilde{p}_{f, \boldsymbol{S}, \boldsymbol{\lambda}, Y, T}(\overline{\boldsymbol{X}}) d\overline{\boldsymbol{X}} = \int_{\mathbb{R}^d} p_{\boldsymbol{\varepsilon}}(\boldsymbol{X} - \overline{\boldsymbol{X}}) \tilde{p}_{\tilde{f}, \tilde{\boldsymbol{S}}, \tilde{\boldsymbol{\lambda}}, \tilde{Y}, \tilde{T}}(\overline{\boldsymbol{X}}) d\overline{\boldsymbol{X}} \\
\Longrightarrow & \left(\tilde{p}_{f, \boldsymbol{S}, \boldsymbol{\lambda}, Y, T} * p_{\boldsymbol{\varepsilon}}\right)(\boldsymbol{X}) = \left(\tilde{p}_{\tilde{f}, \tilde{\boldsymbol{S}}, \tilde{\boldsymbol{\lambda}}, \tilde{Y}, \tilde{T}} * p_{\boldsymbol{\varepsilon}}\right)(\boldsymbol{X}) \\
\stackrel{(c)}{\Longrightarrow} & F\left[\tilde{p}_{f, \boldsymbol{S}, \boldsymbol{\lambda}, Y, T}\right](\boldsymbol{\omega}) \varphi_{\boldsymbol{\varepsilon}}(\boldsymbol{\omega}) = F\left[\tilde{p}_{\tilde{f}, \tilde{\boldsymbol{S}}, \tilde{\boldsymbol{\lambda}}, \tilde{Y}, \tilde{T}}\right](\boldsymbol{\omega}) \varphi_{\boldsymbol{\varepsilon}}(\boldsymbol{\omega}) \\
\stackrel{(d)}{\Longrightarrow} & F\left[\tilde{p}_{f, \boldsymbol{S}, \boldsymbol{\lambda}, Y, T}\right](\boldsymbol{\omega}) = F\left[\tilde{p}_{\tilde{f}, \tilde{\boldsymbol{S}}, \tilde{\boldsymbol{\lambda}}, \tilde{Y}, \tilde{T}}\right](\boldsymbol{\omega}) \\
\Longrightarrow & \tilde{p}_{f, \boldsymbol{S}, \boldsymbol{\lambda}, Y, T}(\boldsymbol{X}) = \tilde{p}_{\tilde{f}, \tilde{\boldsymbol{S}}, \tilde{\boldsymbol{\lambda}}, \tilde{Y}, \tilde{T}}(\boldsymbol{X}).
\end{aligned}
\tag{19}
$$

Step (a) is based on the rule of change-of-variable, where $\operatorname{vol}(\mathbf{A}) = \sqrt{\det\left(\mathbf{A}^T \mathbf{A}\right)}$. In step (b), we define $\tilde{p}_{f, \boldsymbol{S}, \boldsymbol{\lambda}, Y, T}(\boldsymbol{X}) \triangleq p_{\boldsymbol{S}, \boldsymbol{\lambda}}\left(f^{\dagger}(\boldsymbol{X}) \mid Y, T\right) \operatorname{vol}\left(\mathbf{J}_{f^{\dagger}}(\boldsymbol{X})\right) \mathbb{I}_{\mathcal{X}}(\boldsymbol{X})$. In step (c), we use $F[\cdot]$ to denote the Fourier transform. In step (d), we drop $\varphi_{\boldsymbol{\varepsilon}}(\boldsymbol{\omega})$ as it is non-zero *a.e.* (see Assumption 3).

**Step II**. In this step, we transform the equality of the noise-free distributions into the relationship of the sufficient statistics $\boldsymbol{S}$ and $\tilde{\boldsymbol{S}}$. By taking logarithm of both sides of Eq. (19), we have:

$$
\begin{aligned}
& \log \operatorname{vol}\left(J_{f^{\dagger}}(\boldsymbol{X})\right) + \log \mathcal{Q}\left(f^{\dagger}(\boldsymbol{X})\right) - \log \mathcal{C}(Y, T) + \left\langle \boldsymbol{S}\left(f^{\dagger}(\boldsymbol{X})\right), \boldsymbol{\lambda}(Y, T) \right\rangle \\
& = \log \operatorname{vol}\left(J_{\tilde{f}^{\dagger}}(\boldsymbol{X})\right) + \log \tilde{\mathcal{Q}}\left(\tilde{f}^{\dagger}(\boldsymbol{X})\right) - \log \tilde{\mathcal{C}}(Y, T) + \left\langle \tilde{\boldsymbol{S}}\left(\tilde{f}^{\dagger}(\boldsymbol{X})\right), \tilde{\boldsymbol{\lambda}}(Y, T) \right\rangle.
\end{aligned}
\tag{20}
$$

Let $(Y, T)_0, \cdots, (Y, T)_k$ be the $k + 1$ distinct points defined in Assumption 3 - (iv). We obtain $k + 1$ equations by evaluating the Eq. (20) at these points, where the first equation is subtracted from the remaining ones, which leads to the following equation system:

$$
\begin{aligned}
& \left\langle \boldsymbol{S}\left(f^{\dagger}(\boldsymbol{X})\right), \boldsymbol{\lambda}\left((Y, T)_l\right) - \boldsymbol{\lambda}\left((Y, T)_0\right) \right\rangle + \log \frac{\mathcal{C}\left((Y, T)_0\right)}{\mathcal{C}\left((Y, T)_l\right)} \\
& = \left\langle \tilde{\boldsymbol{S}}\left(\tilde{f}^{\dagger}(\boldsymbol{X})\right), \tilde{\boldsymbol{\lambda}}\left((Y, T)_l\right) - \tilde{\boldsymbol{\lambda}}\left((Y, T)_0\right) \right\rangle + \log \frac{\tilde{\mathcal{C}}\left((Y, T)_0\right)}{\tilde{\mathcal{C}}\left((Y, T)_l\right)}, \quad l = 1, \cdots, k.
\end{aligned}
\tag{21}
$$

Let $\mathbf{L}$ be the invertible matrix defined in Assumption 3 - (iv) and $\tilde{\mathbf{L}}$ be the counterpart for $\tilde{\boldsymbol{\lambda}}$, if we summarize all terms irrelevant to $\boldsymbol{X}$ into a constant $\boldsymbol{b}$, we have:

$$
\begin{aligned}
& \mathbf{L}^T \boldsymbol{S}\left(f^{\dagger}(\boldsymbol{X})\right) = \tilde{\mathbf{L}}^T \tilde{\boldsymbol{S}}\left(\tilde{f}^{\dagger}(\boldsymbol{X})\right) + \boldsymbol{b} \\
& \Longrightarrow \boldsymbol{S}\left(f^{\dagger}(\boldsymbol{X})\right) = \mathbf{A} \tilde{\boldsymbol{S}}\left(\tilde{f}^{\dagger}(\boldsymbol{X})\right) + \boldsymbol{c},
\end{aligned}
\tag{22}
$$

where $\mathbf{A} = \mathbf{L}^{-T} \tilde{\mathbf{L}} \in \mathbb{R}^{k \times k}$, and $\boldsymbol{c} = \mathbf{L}^{-T} \boldsymbol{b} \in \mathbb{R}^k$.

**Step III**. Ideally, to prove the element-wise bijective identifiability of the latent variables $\boldsymbol{Z}$, the transformation of the sufficient statistics $\boldsymbol{S}$ derived in Eq. (22) should be bijective. We claim that if the conditional prior $p_{\boldsymbol{S}, \boldsymbol{\lambda}}(\boldsymbol{Z} \mid Y, T)$ is strongly exponential and $\mathbf{L}$ is invertible, $\tilde{\mathbf{L}}$ and $\mathbf{A}$ must also be invertible. The proof is omitted, and can be referred to in Appendix H.1.1 of Lu et al. (2021).

### A.4.2 PART II

In this part, we prove that, if Assumptions 1, 2 and 3 hold, we can identify the factorized part of the sufficient statistics $S(Z)$, i.e., $S_f(Z)$, up to permutation and element-wise transformation. Specifically, if we use $v$ to denote the composite map $\tilde{f}^\dagger \circ f : \mathcal{Z} \to \mathcal{Z}$, Eq. (22) can be rewritten into:

$$S(Z) = \mathbf{A}\tilde{S}(v(Z)) + c. \tag{23}$$

We aim to prove that $\mathbf{A}$ in Eq. (23) is a block permutation matrix.

**Step I**. We start by showing that $v$ is a component-wise function. If we differentiate both sides of Eq. (23) with respect to $Z_s$ and $Z_t$, where $s \neq t$, we have:

$$\frac{\partial S(Z)}{\partial Z_s} = \mathbf{A}\sum_{i=1}^{K_Z} \frac{\partial \tilde{S}(v(Z))}{\partial v_i(Z)} \cdot \frac{\partial v_i(Z)}{\partial Z_s}$$

$$\frac{\partial^2 S(Z)}{\partial Z_s \partial Z_t} = \mathbf{A}\sum_{i=1}^{K_Z}\sum_{i=1}^{K_Z} \frac{\partial^2 \tilde{S}(v(Z))}{\partial v_i(Z)\partial v_j(Z)} \cdot \frac{\partial v_j(Z)}{\partial Z_t} \cdot \frac{\partial v_i(Z)}{\partial Z_s} + \mathbf{A}\sum_{i=1}^{K_Z} \frac{\partial \tilde{S}(v(Z))}{\partial v_i(Z)} \cdot \frac{\partial^2 v_i(Z)}{\partial Z_s \partial Z_t}. \tag{24}$$

Note that for the factorized part of the sufficient statistics $S$, i.e., $S_f$, all *cross-derivatives* are zero, and for the non-factorized part of $S$, i.e., $S_{nf}$, which is a neural network with ReLU activation (i.e., linear *a.e.*), all *second-order derivatives* are zero. Therefore, the *second order cross-derivatives* on the LHS. of Eq. (24) are zero, which leads to the following equality:

$$0 = \mathbf{A}\sum_{i=1}^{K_Z} \frac{\partial^2 \tilde{S}(v(Z))}{\partial v_i(Z)^2} \cdot \frac{\partial v_i(Z)}{\partial Z_t} \cdot \frac{\partial v_i(Z)}{\partial Z_s} + \mathbf{A}\sum_{i=1}^{K_Z} \frac{\partial \tilde{S}(v(Z))}{\partial v_i(Z)} \cdot \frac{\partial^2 v_i(Z)}{\partial Z_s \partial Z_t}. \tag{25}$$

Eq. (25) can be written into the matrix-vector product form as follows:

$$0 = \mathbf{A}\tilde{S}''(Z)v'_{s,t}(Z) + \mathbf{A}\tilde{S}'(Z)v''_{s,t}(Z), \tag{26}$$

where

$$\tilde{S}''(Z) = \left[\frac{\partial^2 \tilde{S}(v(Z))}{\partial v_1(Z)^2}, \cdots, \frac{\partial^2 \tilde{S}(v(Z))}{\partial v_{K_Z}(Z)^2}\right] \in \mathbb{R}^{k \times K_Z},$$

$$v'_{s,t}(Z) = \left[\frac{\partial v_1(Z)}{\partial Z_t} \cdot \frac{\partial v_1(Z)}{\partial Z_s}, \cdots, \frac{\partial v_{K_Z}(Z)}{\partial Z_t} \cdot \frac{\partial v_{K_Z}(Z)}{\partial Z_s}\right]^T \in \mathbb{R}^{K_Z},$$

and

$$\tilde{S}'(Z) = \left[\frac{\partial \tilde{S}(v(Z))}{\partial v_1(Z)}, \cdots, \frac{\partial \tilde{S}(v(Z))}{\partial v_{K_Z}(Z)}\right] \in \mathbb{R}^{k \times K_Z},$$

$$v''_{s,t}(Z) = \left[\frac{\partial^2 v_1(Z)}{\partial Z_s \partial Z_t}, \cdots, \frac{\partial^2 v_{K_Z}(Z)}{\partial Z_s \partial Z_t}\right]^T \in \mathbb{R}^{K_Z}.$$

If we denote the concatenation as $\tilde{S}'''(Z) = \left[\tilde{S}''(Z), \tilde{S}'(Z)\right] \in \mathbb{R}^{k \times 2K_Z}$ and $v''_{s,t}(Z) = \left[v'_{s,t}(Z)^T, v''_{s,t}(Z)^T\right]^T \in \mathbb{R}^{2K_z}$, we have:

$$0 = \mathbf{A}\tilde{S}'''(Z)v'''_{s,t}(Z). \tag{27}$$

Finally, if we denote the rows of $\tilde{S}'''(Z)$ that correspond to the factorized part of $S$ by $\tilde{S}'''_f(Z)$, according to Lemma 5 of (Khemakhem et al., 2020) and the assumption that $k \geq 2K_Z$, we have that the rank of $\tilde{S}'''_f(Z)$ is $2K_Z$. Since $k \geq 2K_Z$, the rank of $\tilde{S}'''_f(Z)$ is also $2K_Z$. Since the rank of $\mathbf{A}$ is $k$, the rank of $\mathbf{A}\tilde{S}'''(Z)$ is $2K_Z$, which implies that $v'''_{s,t}(Z) \in \mathbb{R}^{2K_Z}$ is a zero vector. Therefore, we have $v'_{s,t}(Z) = 0, \forall s \neq t$, and we have demonstrated that $v$ is a component-wise function.

**Step II**. Based on **Step I**, we demonstrate that $\mathbf{A}$ is a block permutation matrix. Without loss of generality, we assume that the permutation in $v$ is Identity, where $v(Z) = [v_1(Z_1), \cdots, v_{K_Z}(Z_{K_Z})]^T$ and each $v_i$ is a nonlinear univariate scalar function. Since $f$ and $\tilde{f}$ are injective, $v$ is bijective and

$v^{-1}(Z) = \left[v_1^{-1}(Z_1), \cdots, v_{K_Z}^{-1}(Z_{K_Z})\right]^T$. If we denote $\overline{S}(v(Z)) = \tilde{S}(v(Z)) + \mathbf{A}^{-1}c$, Eq. (23) can be reformulated as $S(Z) = \mathbf{A}\overline{S}(v(Z))$. We then apply $v^{-1}$ to $Z$ on both sides, which gives

$$S\left(v^{-1}(Z)\right) = \mathbf{A}\overline{S}(Z). \tag{28}$$

Let $t$ be the index of an entry in $S$ that corresponds to the factorized part $S_f$. For all $s \neq t$, we have:

$$0 = \frac{\partial S\left(v^{-1}(Z)\right)_t}{\partial Z_s} = \sum_{j=1}^{k} a_{tj} \frac{\partial \overline{S}(Z)_j}{\partial Z_s}. \tag{29}$$

Since the entries of $\tilde{S}$ are linearly independent, $a_{tj}$ is zero for any $j$ such that $\frac{\partial \overline{S}(Z)_j}{\partial Z_s} \neq 0$. This includes the entries $S_j$ that correspond to (1) the factorized part that does not depend on $Z_t$; and (2) the non-factorized part $S_{nf}$. Therefore, when $t$ is the index of an entry in the sufficient statistics $S$ that corresponds to factor $i$ in the factorized part $S_f$, i.e., $S_{f,i}$, the only non-zero $a_{tj}$ are the ones that map between $S_{f,i}(Z_i)$ and $\overline{S}_{f,i}(v_i(Z_i))$. Therefore, we can construct an invertible submatrix $\mathbf{A}_i'$ with all non-zero elements $a_{tj}$ for all $t$ that corresponds to factor $i$, such that

$$S_{f,i}(Z_i) = \mathbf{A}_i'\overline{S}_{f,i}(v_i(Z_i)) = \mathbf{A}_i'\tilde{S}_{f,i}(v_i(Z_i)) + c_i, \quad i = 1, \cdots, K_Z, \tag{30}$$

where $c_i$ denotes the corresponding elements of $c$. Eq. (30) means that for each $i = 1, \cdots, K_Z$, the matrix block $\mathbf{A}_i'$ of $\mathbf{A}$ affinely transforms the $i$-specific sufficient statistics vector $S_{f,i}(Z_i)$ into $\tilde{S}_{f,i}(v_i(Z_i))$. In addition, there is also an additional block $\mathbf{A}'$ that affinely transforms $S_{nf}(Z)$ in into $S_{nf}(v(Z))$. This completes the proof that $\mathbf{A}$ is a block permutation matrix.

### A.4.3 PART III

Let $\tilde{Z}_i = v_i(Z_i) = \tilde{f}^\dagger(X)_i$ be the $i$th inferred latent variable. Assume again that the permutation in $v$ is Identity. In this part, we prove that if Assumption 2 holds, each inferred latent variable $\tilde{Z}_i$ is the bijective transformation of the true latent variable. The proof is as follows.

*Proof.* Plugging $\tilde{Z}_i$ into Eq. (30), we have:

$$S_{f,i}(Z_i) = \mathbf{A}_i'\bar{S}_{f,i}(\tilde{Z}_i). \tag{31}$$

According to Assumption 2, there exists one dimension of $S_{f,i}$, i.e., $j$, such that $S_{f,ij}$ is bijective. This implies that $S_{f,i}$ is injective, and therefore it has a left-inverse $S_{f,i}^\dagger$. we apply $S_{f,i}^\dagger$ to both sides of Eq. (31), which gives:

$$Z_i = S_{f,i}^\dagger \mathbf{A}_i'\bar{S}_{f,i}(\tilde{Z}_i). \tag{32}$$

Since $\mathbf{A}_i'$ is a block of an invertible block permutation matrix, $\mathbf{A}_i$ is also an invertible matrix, and therefore $\mathbf{A}_i'$ is a bijective mapping. In addition, since $\tilde{S}_{f,i}$ is injective, $\bar{S}_{f,i}$ is also injective, and therefore the composite map $S_{f,i}^\dagger \mathbf{A}_i'\bar{S}_{f,i} : \mathbb{R} \to \mathbb{R}$ that applies on $\tilde{Z}_i$ is a bijective. This completes the proof that each inferred latent variable $\tilde{Z}_i$ is the bijective transformation of the true latent variable in the case of no noise, where $Z = f^\dagger(X)$ are the true latent variables. If noise $\varepsilon$ exists, the posterior distribution of the latent variables can be identified up to an analogous bijective indeterminacy. $\square$

### A.4.4 CONSISTENCY

*Proof.* If the family of the variational posterior $q_\phi(Z|X, Y, T)$ contains the true posterior $p_\theta(Z|X, Y, T)$, then by optimizing the loss of Eq. (9) (with the KL term replaced by the score matching loss defined in Eq. (10)) over its parameter $\phi$, the score matching term will eventually vanish. Therefore, the ELBO term in Eq. (9) will be equal to the log-likelihood. Under this circumstance, CiVAE inherits all the properties of maximum likelihood estimation (MLE). Since the identifiability of CiVAE is guaranteed up to permutation and component-wise bijective transformation of the latent variables, the consistency property of MLE means that the model will converge to the true parameter $\theta^*$ up to such mild indeterminacy of the latent variables in the limit of infinite data. $\square$

## A.5 PROOF OF THEOREM 4.2

*Proof.* Let $C$ be the true latent confounders and $\tilde{C}$ be the transformed confounders, where the transformation function $f$ is bijective and differentiable *a.e.* Let $f^{-1}$ denote its inverse. The ATE estimator that controls transformed confounders $\tilde{C}$ can be formulated as:

$$DEV(\tilde{C}) = \mathbb{E}_{p(\tilde{C})}[\mathbb{E}[Y|T=1, \tilde{C}=\tilde{c}] - \mathbb{E}[Y|T=0, \tilde{C}=\tilde{c}]]. \tag{33}$$

Specifically, for the continuous case where density functions exist, for each term, we have:

$$\mathbb{E}_{p(\tilde{C})}[\mathbb{E}[Y|T=t, \tilde{C}=\tilde{c}]] = \int f_{\tilde{C}}(\tilde{c}) \int y \cdot f_{Y|T,\tilde{C}}(y|t, \tilde{c}) dy d\tilde{c}. \tag{34}$$

For the marginal density $f_{\tilde{C}}(\tilde{c})$, the following equality holds:

$$f_{\tilde{C}}(\tilde{c}) = f_C(f^{-1}(\tilde{c}))|J_{f^{-1}}(\tilde{c})| = f_C(c)|J_{f^{-1}}(\tilde{c})|. \tag{35}$$

As for the conditional density $f_{Y|T,\tilde{C}}(y|t, \tilde{c})$, since $f$ is bijective, according to Eq. (12), we have:

$$f_{Y|T,\tilde{C}}(y|t, \tilde{c}) = f_{Y|T,C}(y|t, c). \tag{36}$$

Combining Eqs. (35) and (36), and given that $d\tilde{c} = |J_f(c)|dc$, we have:

$$\begin{aligned}
(34) &= \int f_C(c)|\mathbf{J}_{f^{-1}}(\tilde{c})| \int y \cdot f_{Y|T,C}(y|t, c) dy |\mathbf{J}_f(c)| dc \\
&= |\mathbf{J}_{f^{-1}}(\tilde{c})| \cdot |\mathbf{J}_f(c)| \int f_C(c) \int y \cdot f_{Y|T,C}(y|t, c) dy dc \\
&\overset{(a)}{=} \int f_C(c) \int y \cdot f_{Y|T,C}(y|t, c) dy dc \\
&= \mathbb{E}_{p(C)}[\mathbb{E}[Y|T=t, C=c]],
\end{aligned} \tag{37}$$

where the term $|J_{f^{-1}}(\tilde{c})| \cdot |J_f(c)|$ vanishes in step (a) as the two factors have the product of one. Therefore, if we plug Eq. (37) into Eq. (33), it leads to the following equality:

$$\begin{aligned}
DEV(\tilde{C}) &= \mathbb{E}_{p(\tilde{C})}[\mathbb{E}[Y|T=1, \tilde{C}=\tilde{c}] - \mathbb{E}[Y|T=0, \tilde{C}=\tilde{c}]] \\
&= \mathbb{E}_{p(C)}[\mathbb{E}[Y|T=1, C=c] - \mathbb{E}[Y|T=0, C=c]] = DEV(C) = ATE,
\end{aligned} \tag{38}$$

where the last step is due to Eq. (2) in Definition 2, which completes our proof that controlling bijectively transformed confounders provides an unbiased estimation of ATE. $\square$

## B RELATED WORK

### B.1 POST-TREATMENT BIAS IN CAUSAL INFERENCE

Bias due to accidentally controlling post-treatment variables, i.e., *post-treatment bias*, has long been recognized as dangerous in causal effect estimation (King, 2010). Back at 2005, Pearl (2015) cautioned that controlling more is not better, and uses the collider bias (Elwert & Winship, 2014) and M-Bias (Ding & Miratrix, 2015) as two examples to show that bias can be increased when controlling the post-treatment variables. Furthermore, Montgomery et al. (2018) show that indirect correlations between post-treatment variable $M$ and outcome $Y$ can still cause bias. Recent works prove that even if $M$ has no causal relationship with $Y$, controlling it can still increase the variance of estimand (Hassanpour & Greiner, 2020). However, most of these works study the post-treatment bias in the observational space, where latent post-treatment variables that are mixed with confounders to generate the observed covariates can be easily ignored by the researcher. Therefore, it motivates us to develop CiVAE, which is robust to the latent post-treatment bias under mild assumptions.

### B.2 COVARIATE DISENTANGLEMENT

Recently, researchers have realized that directly controlling the proxy of confounders $\mathbf{X}$ may not be safe, as variables other than confounders could lurk in the confounder proxy and ruin the ATE

estimation (Hassanpour & Greiner, 2020). Traditional methods assume that the variables that generate the observed covariates $\mathbf{X}$ are a mixture of confounders, adjusters, and influencers (Shalit et al., 2017), where adjusters should not be controlled as it can increase the estimation variance (Hassanpour & Greiner, 2019). Most methods rely on the fact that adjusters are correlated only with the treatment to separate them from other variables (Hassanpour & Greiner, 2020; Zhang et al., 2021) (see Fig. (2)). This can also be used to remove post-treatment variables that are not correlated with the outcome, which have similar statistical properties with adjusters (Wang et al., 2023). Here, a different work is NICE (Shi et al., 2021), which uses the fact that confounders and influencers are direct causal parents of the outcome to find these variables with invariant learning as the control set (Arjovsky et al., 2019). However, since mediators are also direct parents of the outcome, NICE is still not robust to general post-treatment bias. Given that all the above methods cannot satisfactorily address the latent post-treatment in general cases, it is imperative to design the CiVAE, where confounders can be identified and disentangled with latent post-treatment variables for unbiased adjustment. We note that Xu et al. propose CFDiVAE that identifies sufficient mediators from covariates with iVAE and estimates the ATE via front-door adjustment, which is an interesting alternative to CiVAE provided that the covariates $X$ contain all mediators between $T$ and $Y$ where front-door criterion satisfies.

## C  EXTENDING CiVAE TO ADDRESS LATENT INTERACTIONS

### C.1  EXAMPLES FOR THE IDENTIFICATION CRITERIA

In this section, we provide concrete examples to more intuitively demonstrate the ability of CiVAE to address (non-cyclic) interactions among the latent confounders $\boldsymbol{C}$ and latent post-treatment variables $\boldsymbol{M}$ introduced in Section 4.5, which are discussed as follows:

***Example 1***. *Intra-Confounder Interactions*.

In this example, latent confounders are allowed to interact with each other, i.e., for arbitrary $i'$, $j'$, $C_{i'}$ could be the causal parent of $C_{j'}$. In this example, if two inferred latent variables $\hat{Z}_i, \hat{Z}_j \in \hat{\boldsymbol{Z}}$ (which individually identify the true latent variables according to Theorem 4.1) are latent confounders, i.e., $\hat{C}_{i'}, \hat{C}_{j'}$ (***case 1****), we have $\hat{\boldsymbol{Z}}_C = PA(\hat{C}_{i'}) \cap PA(\hat{C}_{j'}) \subset \hat{\boldsymbol{C}}$ , where $PA$ denotes the parent set of the node in the true causal graph. For ***cases 2****,***3****, we have $\hat{\boldsymbol{Z}}_M = \hat{\boldsymbol{Z}}_{C,M} = \emptyset$.

***Example 2***. *Intra-Post-Treatment Interactions*.

In this example, post-treatment variables are allowed to sequentially (i.e., non-cyclic) influence one another, i.e., for arbitrary $i'$, $j'$, $M_{i'}$ could be the causal parents of $M_{j'}$ if no circle is formed. In this case, if two inferred latent variables $\hat{Z}_i, \hat{Z}_j \in \hat{\boldsymbol{Z}}$ (which individually identify the true latent variables according to Theorem 4.1) are latent post-treatment variables, i.e., $\hat{M}_{i'}, \hat{M}_{j'}$ (***case 2****), we have $\hat{\boldsymbol{Z}}_M = PA(\hat{M}_{i'}) \cap PA(\hat{M}_{j'}) \subset \hat{\boldsymbol{M}}$. For ***cases 1****,***3****, we have $\hat{\boldsymbol{Z}}_C = \hat{\boldsymbol{Z}}_{C,M} = \emptyset$.

***Example 3***. *Cross Confounder-Post-treatment Variable Interactions*.

In this example, confounders are allowed to influence the post-treatment variables (note that since confounders are pre-treatment, they cannot be influenced by the post-treatment variables). In this example, similar to ***Example 2***, if two inferred latent variables $\hat{Z}_i, \hat{Z}_j \in \hat{\mathbf{Z}}$ (which individually identify the true latent variable) are latent post-treatment variables, i.e., $\hat{M}_{i'}, \hat{M}_{j'}$ (***case 2****), we have $\hat{\boldsymbol{Z}}_M = PA(\hat{M}_{i'}) \cap PA(\hat{M}_{j'}) \subset \hat{\boldsymbol{C}}$. For ***cases 1****,***3****, we have $\hat{\boldsymbol{Z}}_C = \hat{\boldsymbol{Z}}_{C,M} = \emptyset$.

More complicated cases can be viewed as combinations of Examples 1, 2, and 3, where the condition sets $\hat{\boldsymbol{Z}}_C, \hat{\boldsymbol{Z}}_M, \hat{\boldsymbol{Z}}_{C,M} = \emptyset$ can be directly derived by utilizing the above conclusions.

### C.2  ADDITIONAL EXPERIMENTS[3]

#### C.2.1  INTRA-INTERACTIONS AMONG LATENT MEDIATORS

In this subsection, we empirically analyze the case where latent post-treatment variables $\boldsymbol{M}$ interact with each other. Specifically, we extend the simulated datasets described in Section 5.1, where we

---

[3]Code available at `https://github.com/yaochenzhu/CiVAE`.

Table 2: Comparison of CiVAE with baselines when intra-interactions among $M$ exist.

| Dataset | LatentMediator | | LatentCorrelator | | LinkedIn(Age) | | LinkedIn(Gender) | |
|---|---|---|---|---|---|---|---|---|
| Method | ATE. | Err. | ATE. | Err. | ATE. | Err. | ATE. | Err. |
| CEVAE | $1.627 \pm 0.549$ | 2.627 | $2.659 \pm 0.302$ | 1.353 | $0.152 \pm 0.027$ | 0.420 | $-0.225 \pm 0.044$ | -0.144 |
| TEDVAE | $1.653 \pm 0.511$ | 2.042 | $2.827 \pm 0.259$ | 1.521 | $0.180 \pm 0.047$ | 0.448 | $-0.189 \pm 0.012$ | -0.108 |
| CiVAE | $\mathbf{-0.350} \pm 0.695$ | **1.785** | $\mathbf{1.785} \pm 0.481$ | **0.479** | $\mathbf{-0.073} \pm 0.101$ | **0.195** | $\mathbf{-0.136} \pm 0.087$ | **-0.055** |
| True ATE | $-1.000 \pm 0.000$ | 0.000 | $1.306 \pm 0.000$ | 0.000 | $-0.268 \pm 0.000$ | 0.000 | $-0.081 \pm 0.000$ | 0.000 |

Table 3: Comparison of CiVAE with baselines when inter-interactions between $C$ and $M$ exist.

| Dataset | LatentMediator | | LatentCorrelator | | LinkedIn(Age) | | LinkedIn(Gender) | |
|---|---|---|---|---|---|---|---|---|
| Method | ATE. | Err. | ATE. | Err. | ATE. | Err. | ATE. | Err. |
| CEVAE | $2.070 \pm 0.279$ | 3.070 | $2.831 \pm 0.398$ | 1.831 | $0.094 \pm 0.061$ | 0.362 | $-0.192 \pm 0.015$ | -0.111 |
| TEDVAE | $1.743 \pm 0.307$ | 2.743 | $2.954 \pm 0.763$ | 1.954 | $0.109 \pm 0.116$ | 0.377 | $-0.212 \pm 0.019$ | -0.131 |
| CiVAE | $\mathbf{-0.716} \pm 0.523$ | **0.284** | $\mathbf{1.385} \pm 0.660$ | **0.385** | $\mathbf{-0.041} \pm 0.144$ | **0.227** | $\mathbf{-0.129} \pm 0.064$ | **-0.048** |
| True ATE | $-1.000 \pm 0.000$ | 0.000 | $1.000 \pm 0.000$ | 0.000 | $-0.268 \pm 0.000$ | 0.000 | $-0.081 \pm 0.000$ | 0.000 |

make *(i)* $T$ directly affects $M_1$, *(ii)* $M_1$ affects $M_2$, and *(iii)* $M_1$, $M_2$ affect $M_3$. The coefficients are randomly sampled from $\mathcal{N}(0, 1/3)$. The results in Table 2 demonstrate that the adapted CiVAE is still more robust to latent post-treatment bias compared to CEVAE and TEDVAE.

### C.2.2 Inter-Interactions between Latent Mediators and Latent Confounders

In this subsection, we empirically analyze another case where inter-interactions exist between latent confounders $C$ and latent post-treatment variables $M$. Specifically, we extend the simulated datasets described in Section 5.1 to allow each latent confounder $C_i \in \mathbb{R}^3$ to determine $M \in \mathbb{R}^3$. The coefficients are randomly sampled from $\mathcal{N}(0, 1/3)$. The results in Table 3 demonstrate that the adapted CiVAE is still significantly more robust to latent post-treatment bias compared to CEVAE and TEDVAE, which empirically verify our claim that the adapted CiVAE can address the case where inter-interactions exist among latent confounders and post-treatment variables.

### C.3 Robustness Analysis of CiVAE

### C.3 Robustness Analysis of CiVAE

### C.3.1 Sensitivity w.r.t. Assumption 1

In this section, we evaluate the sensitivity of CiVAE to the injectivity assumption stated in Assumption 1. For the `latentMediator` and `latentCorrelator` cases outlined in Section 5.1, the mixture function $f$ corresponds to the matrix $\mathbf{A}$ described in Corollaries 3.3 and A.1, respectively. When $K_X \geq K_C + K_M$, directly random sampling of $\mathbf{A}$ results in $\mathbf{A}$ being almost surely injective.

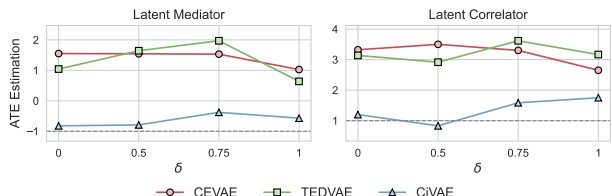

Figure 6: Sensitivity of CiVAE to injectivity assumption.

To make $\mathbf{A}$ approach non-injectivity, we first perform singular value decomposition (SVD) on $\mathbf{A}$ yielding $\mathbf{U}^T \mathbf{\Lambda} \mathbf{V}$, where $\mathbf{U}$ and $\mathbf{V}$ are orthogonal matrices of left/right eigenvectors, and $\mathbf{\Lambda}$ is the diagonal matrix of singular values. We then dampen the largest singular value in $\mathbf{\Lambda}$ by a factor of $1 - \delta$, resulting in the modified singular value matrix $\mathbf{\Lambda}_{1-\delta}$. The dampened reconstruction $\mathbf{A}_{1-\delta} = \mathbf{U}^T \mathbf{\Lambda}_{1-\delta} \mathbf{V}$ is used to generate the covariates $X$ from $C$, $M$ instead of $\mathbf{A}$. When $\delta = 0$, $f = \mathbf{A}_1$ maintains injectivity. As $\delta$ approaches 1, $f = \mathbf{A}_{1-\delta}$ accordingly approaches non-injectivity.

The evaluation of CiVAE, CEVAE, and TEDVAE on data generated under different $\delta$ are illustrated in Fig. 6. From Fig. 6, we observe that violation of injectivity assumption indeed has a negative influence on the accuracy of ATE estimation. However, even with the decrease of $\delta$, which indicates $f$ approaching non-injectivity, CiVAE is still consistently more robust to latent post-treatment bias and outperforms both CEVAE and TEDVAE in terms of maintaining accurate ATE.

### C.3.2 SENSITIVITY W.R.T. LATENT DIMENSION

In this section, we explore the sensitivity of CiVAE to model misspecification. Specifically, we vary the dimension of latent variables assumed by CiVAE, i.e., $Dim(\hat{\boldsymbol{Z}})$, to values in $[4, 6, 8, 10]$ for both the `latentMediator` and the `latentCorrelator` cases outlined in Section 5.1 (with the correct latent dimension being 6) and compare its performance with CEVAE and TEDVAE. The results are illustrated in Fig. 7. In Fig. 7, the upper figures show the cases where no interactions exist among latent variables, and the lower figures show the cases

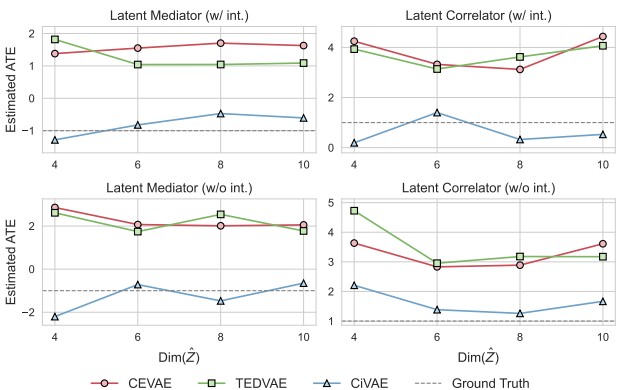

Figure 7: Sensitivity of CiVAE to model misspecification.

when latent confounders also confound the latent post-treatment variables (see Section C.2.2). From Fig. 7, we find that underestimating the latent dimension does have an evident negative influence on CiVAE. However, CiVAE demonstrates good robustness to the overestimate in the latent dimension, showing more robustness of latent post-treatment bias to CEVAE and TEDVAE even when the dimension is mis-specified. This highlights CiVAE's ability to adapt to model specifications and underscores its enhanced robustness to latent post-treatment bias compared to the baseline methods.

