# OpenReview forum: "Causal Effect Estimation with Mixed Latent Confounders and Post-treatment Variables"
_ICLR.cc/2025/Conference — ICLR 2025 Poster_

### Official Review · Reviewer_JSgY · 2024-10-20

**Soundness:** 2
**Presentation:** 3
**Contribution:** 2
**Rating:** 6
**Confidence:** 3

**Summary:**

The paper introduces the Confounder-identifiable Variational Autoencoder (CiVAE) to estimate treatment effects in settings where the observed covariates are a function of latent confounders and latent post-treatment variables. The proposed method is designed to disentangle these latent components, and it is provably unbiased under some specific assumptions. Through experiments on both simulated and real-world datasets, the method demonstrates superior performance compared to existing baselines.

**Strengths:**

- The issue of post-treatment bias is understudied, making this paper a valuable contribution to the field.

- The paper is well-written, which makes it easy to understand the core ideas.

- The empirical validation shows that the method outperforms existing baselines.

**Weaknesses:**

- **Main concern**. The method depends on strong and untestable assumptions for identifiability, e.g. the injective mapping described in Assumption 1. Such assumptions may be difficult to verify, even with domain knowledge, and fall short of the causal inference field's emphasis on interpretable and justifiable identification assumptions.
- **Other concerns**. While post-treatment variables are recognized as an issue, I am skeptic regarding the significance of *latent* post-treatment variables as a problem. The example provided does not clarify why these latent variables pose a significant issue, and I'm uncertain about its practical relevance.
- **Other concerns**. The assumption of independence between latent confounders and latent post-treatment variables (see Fig. 1c) is quite strong and unlikely to hold in any realistic application.

**Questions:**

- Could the author provide some examples in the causal inference literature that involve the use of post-treatment proxy variables?
- Under what circumstances do you expect the injectivity assumption to hold, given that even some simple linear models may fail to satisfy this?
- Can the author share experimental results showing the impact of violating injectivity in Assumption 1?

---

> ### Author Response · Authors · 2024-11-18
> **Official Response of Submission5496 by Authors - Part 1**
>
> Thank you very much for your constructive feedback. It's our pleasure to have this valuable opportunity to discuss your concerns/questions with you. We believe that our paper will be much stronger thanks to your efforts.
>
> >  The method depends on strong and untestable assumptions for identifiability, e.g. the injective mapping described in Assumption 1. Such assumptions may be difficult to verify, even with domain knowledge, and fall short of the causal inference field's emphasis on interpretable and justifiable identification assumptions.
>
> For Assumption 1, we note that when latent dimension $dim(C) + dim(M)$ is less than the dimension of the observed covariates $dim(X)$, non-injective maps from $[C, M]$ to $X$ have measure zero in the functional space $\{f: \{C, M\} \rightarrow X\}$, i.e., Assumption 1 almost surely holds. Therefore, we wouldn't call the assumption strong.
>
> Additionally, domain knowledge can assist in selecting $X$, as Assumption 1 encourages the inclusion of more covariates that contain different views on the latent confounders, even if some selected covariates contain post-treatment components. This actually weakened the strong and untestable ignorability assumption made by existing proxy-based methods.
>
> Furthermore, in another iVAE-based method that infers front-door variables from observed covariates published in ICLR'24 suggested by reviewer 22EG, injectivity is also assumed (see the paragraph under Eq. (10))
>
> However, we agree that it is important to analyze the sensitivity of CiVAE as $f$ approaches non-injectivity. Based on your suggestion, we have included experiments that gradually nudge $f$ towards non-injectivity (by dampening the largest singular values to zero) in Section C.3.1 of the Appendix, where we find that CiVAE is still consistently more robust to latent post-treatment bias and outperforms the baselines in terms of maintaining accurate ATE. Thank you for this constructive comment.
>
> > While post-treatment variables are recognized as an issue, I am skeptic regarding the significance of _latent_ post-treatment variables as a problem.
>
> In observational studies where data for ATE estimation are collected *post hoc*, it is quite common for observed covariates to be generated from an entanglement of pre-treatment and post-treatment components, which poses a significant challenge for causal effect estimation. In our paper, we have provided several examples of latent post-treatment variables in causal inference across fields such as social science (Lines 73-88), medicine (Lines 90-95), and politics and economics (Lines 95-101). We have further clarified these points by adding Lines 90-91 to the revised manuscript.
>
> > The assumption of independence between latent confounders and latent post-treatment variables (see Fig. 1c) is quite strong and unlikely to hold in any realistic application.
>
> We appreciate your concern. In fact, we have discussed the generalization of CiVAE to addressing arbitary interactions among the latent variables $[C, M]$ and provided the proof for identification in Section 4.5 in the main paper, with corresponding experiments provided in Section C in the Appendix to demonstrate the effectiveness of the proposed strategy.
>
> Here, we sincerely apologize the confusion caused by Fig. 2-(c). With the generalization strategy discussed in  Section 4.5, CiVAE allows arbitary dependence among latent variables for confounder identification and unbiased ATE estimation. To avoid future confusion, we have clarified this by adding dashed arrows in Fig. 2-(c) to represent potential dependencies among latent variables in the revised manuscript.
>
> [1] Causal Inference with Conditional Front-Door Adjustment and Identifiable Variational Autoencoder, ICLR 2024.

---

> ### Author Response · Authors · 2024-11-18
> **Official Response of Submission5496 by Authors - Part 2**
>
> > Could the author provide some examples in the causal inference literature that involve the use of post-treatment proxy variables?
>
> Thank you for the constructive feedback! Most papers on causal effect estimation typically assume away post-treatment variables by relying on the strong ignorability assumption, where the proxy is assumed to consist solely of pre-treatment variables.
>
> However, in the field of causal mediation analysis, there are some papers [2,3] that explore the use of proxies for post-treatment variables, particularly when the mediator cannot be directly observed, but they are not very relevant to our setting. Additionally, Reviewer 22EG brought to our attention a paper [1]  that uses iVAE to identify latent mediators from covariates and uses front-door adjustment for ATE estimation. This approach relies on a completely different set of untestable assumptions, such as $X$ containing all mediators that block the path between $T$ and $Y$, for identifiability. This could be an interesting alternative to CiVAE. We have discussed this work in Section B.2 as per Reviewer 22EG's request.
>
> > Under what circumstances do you expect the injectivity assumption to hold, given that even some simple linear models may fail to satisfy this?
>
>
> Thank you for your question. As we clarified in our previous response, when $\text{dim}(C) + \text{dim}(M) < \text{dim}(X)$, non-injective maps from $[C, M]$ to $X$ have measure zero in the functional space ${f: {C, M} \rightarrow X}$, meaning Assumption 1 almost surely holds.
>
> Regarding the counterexample of "simple linear models may fail to satisfy this," it's important to note that this issue arises only when $\text{dim}(C) + \text{dim}(M) > \text{dim}(X)$. For example, with one-dimensional (1D) $X$, 1D $C$, and 1D $M$, a simple linear transformation like $X = C + M$ indeed violates the injective assumption. However, if we have a two-dimensional (2D) $X = [X_{1}, X_{2}]$, even if $X_{1}$ and $X_{2}$ are both simple linear transformations of $C$ and $M$, such as $X_{1} = C + M$ and $X_{2} = C - M$, the mapping $f: [C, M] \rightarrow X$ is injective almost surely. Therefore, to ensure the injectivity assumption holds, we can include more variables in $X$ that are assumed to contain confounders. We hope this intuitive explanation with examples further clarifies your concerns regarding Assumption 1.
>
> > Can the author share experimental results showing the impact of violating injectivity in Assumption 1?
>
> We have included the experiments in Section C.3.2 of the revised manuscript. Thank you very much for your constructive feedback.
>
> [2] The mediation formula: A guide to the assessment of causal pathways in nonlinear models.
> [3] Bayesian causal mediation analysis with latent mediators and survival outcome.

---

> > ### Comment · Reviewer_JSgY · 2024-11-18
> >
> > Thanks for the addressing my concerns. I have adjusted my score accordingly.

---

> > > ### Author Response · Authors · 2024-11-18
> > > **Thank You for the Acknowledgment**
> > >
> > > Dear reviewer JSgY,
> > >
> > > We sincerely appreciate your thoughtful feedback and thank you again for your time and efforts in the review and discussion phases.
> > >
> > > Best,
> > > Authors

---

### Official Review · Reviewer_22EG · 2024-11-02

**Soundness:** 3
**Presentation:** 3
**Contribution:** 3
**Rating:** 8
**Confidence:** 4

**Summary:**

This paper addresses the challenge of handling latent confounders and post-treatment variables. Recent research has attempted to solve this problem by using proxy variables. However, selected proxies may inadvertently combine both confounders and post-treatment variables, potentially introducing bias into estimates. To tackle this, the authors propose a novel method called the Confounder-identifiable Variational Autoencoder (CiVAE) to mitigate this bias.

**Strengths:**

1.I find the problem they studied intriguing, specifically how proxy attributes may cover both confounders and post-treatment variables. It is crucial to separate these biases throughout the causal effect estimation process.

2. The paper is well-written and theoretically sound, with proofs for all statements provided in the appendix.

3.The core idea is clearly explained and easy to follow. Including a code link improves reproducibility.

**Weaknesses:**

1. In Figure 2, there is a bi-directional path between M and Y. However, in the following example, M is treated as a strict mediator. Please ensure consistency in the directions of M. Or add more examples to show if the case T->M<-Y happened. In line 269, it states, "However, since both C and M form fork structures with the outcome Y (see Fig. 2-(c))." I am wondering why M forms fork structures.

2. Z is the true latent space (Z = [C, M]). According to the paper, there exists a bi-directional path between M and Y. Could you explain why only p(Z|Y,T) is considered? If T → M → Y, I agree with using p(Z|T); if T → M ← Y, I agree with p(Z|T,Y). I suggest considering different cases using different forms of p(·).

3. More interestingly, if Z also includes C, then is C identifiable? I am wondering if C is treated as a general case in CEVAE without considering identifiability. Even with iVAE, additional information may be missing to guide identifiability.

4. A related work may need to be included. [1] discusses the iVAE learning process of the front-door adjustment set, which is a very similar setting to this paper. I suggest adding a discussion of this in the related work section.

[1] Causal Inference with Conditional Front-Door Adjustment and Identifiable Variational Autoencoder in ICLR 2024.

**Questions:**

See Above.

---

> ### Author Response · Authors · 2024-11-19
> **Official Response of Submission5496 by Authors - Part 1**
>
> Thank you very much for your constructive feedback. It's our pleasure to have this valuable opportunity to discuss your concerns/questions with you. We believe that our paper will be much stronger thanks to your efforts.
>
> >  In Figure 2, there is a bi-directional path between M and Y. However, in the following example, M is treated as a strict mediator. Please ensure consistency in the directions of M. Or add more examples to show if the case T->M<-Y happened.
>
> We apologize for the confusion. The bi-directional dashed paths between $M$ and $Y$ in Fig. 2-(c) indicate that we allow arbitrary correlations $M$ and $Y$. The identifiability of CiVAE holds regardless of the specific relationship between $M$ and $Y$. We have clarified this point in the caption of Fig. 2.
>
> Additionally, beyond the latent mediator scenario discussed in Corollary 1, we also consider another case in the paper, referred to as "LatentCorrelator," where $M$ and $Y$ are confounded by another latent vector $U$. The theoretical analysis for this case is provided in Corollary A.1 in Section A of the Appendix, and the corresponding experimental results are presented in the "LatentCorrelator" column of Tables 1, 2, and 3.
>
> > In line 269, it states, "However, since both C and M form fork structures with the outcome Y (see Fig. 2-(c))." I am wondering why M forms fork structures.
>
> Thank you for pointing out this typo. We have revised it to "However, since $C$ form fork structures with and *$M$ could have arbitrary relations with $Y$*" in the revised manuscript. The remaining parts are not affected by the typo, as our intention is to show that latent variables in $[C, M]$ are not independent given $Y$, where the assumptions of naive iVAE do not hold.
>
> >  Z is the true latent space (Z = [C, M]). According to the paper, there exists a bi-directional path between M and Y. Could you explain why only p(Z|Y,T) is considered? If T → M → Y, I agree with using p(Z|T); if T → M ← Y, I agree with p(Z|T,Y). I suggest considering different cases using different forms of p(·).
>
> We apologize for the confusion caused by Eq. (8). To clarify, it's important to distinguish between (i) structural equations and (ii) conditional distributions that are measurable in the data. In terms of  _structural equations_, for the case $T \to M \to Y$, we indeed have $p_{se}(Z|T)$, and for $T \to M \leftarrow Y$, we have $p_{se}(Z|T, Y)$. However, $p_{\theta}(X, Z \mid Y, T)$ in Eq.  (8) refers to the  _conditional distribution_  of $X$ and $Z$ given $Y$ and $T$.
>
> The factorization in the generative step of Eq. (8), i.e., $p_{\theta}(X, Z \mid Y, T) = p_{f}(X \mid Z) \cdot p_{S, \lambda}(Z \mid Y, T)$, is based on the law of total probability and the conditional independence assumptions implied by the causal graph in Fig. 2-(c). Here, $Y$ and $T$ are omitted from the conditions in $p_{f}(X \mid Z)$ because $Z$ is the Markov blanket of $X$. We have clarified these points in Lines 307-308 of the revised manuscript.
>
> >  More interestingly, if Z also includes C, then is C identifiable? I am wondering if C is treated as a general case in CEVAE without considering identifiability. Even with iVAE, additional information may be missing to guide identifiability.
>
> Thanks for raising this important question. In our case, latent variables $Z$ that generate the observed covariates $X$ include both $C$ (latent confounders) and $M$ (mixed-in latent post-treatment variables). iVAE itself is indeed not enough for the identification. However, the key contribution of CiVAE is to extend iVAE by ***(i)*** allowing identification of $Z$ with arbitrary conditional dependence of $Z$ on $T, Y$, ***(ii)*** allowing disentanglement of $C$ from $M$ in $Z$ even if arbitrary interactions exist among $C$ and $M$, and ***(iii)*** ensuring the component-wise bijective transformation of inferred $\hat{C}_{i}$ to avoid collapse of latent confounders.
>
> Intuitively,  ***(i)*** and ***(iii)*** are achieved by adding a very weak assumption on iVAE that the conditional distribution $p(Z|T,Y)$ belongs to the exponential family where the non-factorized part of sufficient statistics can be parameterized by ReLU neural networks, and the factorized part has at least one invertible dimension (intuition on why this works see remarks under Assumption 2). The assumption is weak as ReLU NNs have universal approximation ability, and most commonly used exponential family distributions have invertible dimensions in sufficient statistics. ***(ii)*** is achieved by utilizing the invariant causal relations among $C$, $M$, $T$. The details can be referred to in Section 4.4 of the main paper.

---

> ### Author Response · Authors · 2024-11-19
> **Official Response of Submission5496 by Authors - Part 2**
>
> > A related work may need to be included. [1] discusses the iVAE learning process of the front-door adjustment set, which is a very similar setting to this paper. I suggest adding a discussion of this in the related work section.
>
> Thank you for bringing this important work to our attention! We find [1] particularly interesting as it employs iVAE to identify latent mediators from covariates and utilizes front-door adjustment for unbiased ATE estimation. We have included a detailed discussion of [1] in Section B.2 of the revised manuscript.

---

### Official Review · Reviewer_PTBp · 2024-11-03

**Soundness:** 2
**Presentation:** 2
**Contribution:** 3
**Rating:** 6
**Confidence:** 4

**Summary:**

The manuscript proposes CiVAE method to address the risk of conditioning on post-treatment (latent) variables in estimating causal effects, by disentangling the latent unobserved confounders from the latent post-treatment variables. The identification result of CiVAE and identification of latent confounders are shown. The authors also show the empirical results by a simulated data and a real data.

**Strengths:**

- The authors study a important issue of latent post-treatment bias.
- Pratical algorithm and sound theoretical results are provided under certain conditions.
- Empirical performance is strong comparing with existing ones.

**Weaknesses:**

- Assumption 1 requires $f$ to be injective, which is much stronger than common conditions in the literature, even the dimension of the observation space is larger than dim(C) + dim(M). In many cases, there might be a  latent confounder or post-treatment variable not correlated with $X$. It is suggested to check if the proposed method sensitive to the injective condition.

- The interactions among latent variables might come from other unobserved variables that confound $Z_i$ and $Z_j$, not belonging to three cases in section 4.5.

- How to determine the dimension of Z? How's the different selections of dim(Z) affect your ATE and identification of latent confounders with or without interactions?

- Please fix the typo in equation (8).

**Questions:**

See weaknesses

---

> ### Author Response · Authors · 2024-11-18
> **Official Response of Submission5496 by Authors**
>
> Thank you very much for your constructive feedback. It's our pleasure to have this valuable chance to discuss with you and address your concerns/questions. We believe that our paper will be much stronger thanks to your efforts.
>
> >  In Assumption 1, in many cases, there might be a latent confounder or post-treatment variable not correlated with X.
>
> Since our goal is to control *only* latent confounder component in the observed covariates $X$ to avoid post-treatment bias in ATE estimation, we are only interested in the **mixed-in** latent post-treatment variables that causally determines $X$ (see Fig. 2-(c)). It is acceptable if other latent post-treatment variables are omitted, as this is actually the desired case! We apologize for the confusion and have revised Lines 145 and 149 in Section 3.1 to clarify this point.
>
> In addition, covariates $X$ containing all latent confounders are assumed by most proxy-based methods via the strong ignorabiliy assumption. Therefore, we do not introduce extra assumptions compared with existing literature (we actually make it weaker by allowing latent post-treatment variables to be mixed in $X$)
>
> >  Assumption 1 requires f to be injective, which is much stronger than common conditions.
>
> After clarifying the first question, we note that when $dim(C) + dim(M) < dim(X)$, the non-injective maps from $[C, M]$ to $X$ have measure zero in the functional space {$\{f: \{C, M\} \rightarrow X\}$}, i.e., the assumption *almost surely holds*. Therefore, we wouldn't call the assumption strong. Furthermore, injectivity is also assumed in another iVAE-based method that infers front door variables from observed covariates published in ICLR'24 [1] suggested by reviewer 22EG (see the paragraph under Eq. (10)) .
>
> However, we agree with the reviewer that it is important to analyze the sensitivity of CiVAE as $f$ approaches non-injectivity. Based on your suggestion, we have included experiments where we gradually nudge $f$ towards non-injectivity (by dampening the largest singular values to zero) in Section C.3.1 of the Appendix, where we find that CiVAE is still consistently more robust to latent post-treatment bias and outperforms the baselines in terms of maintaining accurate ATE. Thank you for this constructive feedback.
>
> >  The interactions among latent variables might come from other unobserved variables that confound  Z_{i}  and  Z{j}, not belonging to three cases in section 4.5.
>
> Thank you for pointing this out. We should indeed clarify that to address latent interactions, CiVAE requires the assumption that "there are no other unobserved confounders beyond $C$ that confound the latent variables $Z$." We have added this clarification to Lines 135-136 of the revised manuscript.
>
> However, the three cases outlined in Section 4.5 are sufficiently general. Once unobserved latent confounders over latent variables are excluded, CiVAE can individually identify $C$ mixed in $X$, even in the presence of arbitrary interactions among $C$ and $M$.
>
> > How to determine the dimension of Z? How's the different selections of dim(Z) affect your ATE and identification of latent confounders with or without interactions?
>
> Thank you for raising this important question. CiVAE, like CEVAE, empirically sets the dimension of latent variables based on prior judgment. It was indeed our neglect not to test the model's robustness with respect to dimension mis-specification. Based on your suggestion, we have included experiments that varies the latent dimension assumed by CiVAE in Section C.3.2 of the Appendix. These experiments demonstrate that CiVAE maintains better robustness compared to other proxy-based methods, even when the dimension of $\hat{Z}$ is mis-specified.
>
> > Please fix the typo in equation (8).
>
> There is indeed a typo in Step $(i)$ of Eq. (8). We have corrected it in the revised manuscript, and it should read $\ p_{\theta}(X, Z \mid Y, T) = p_{f}(X \mid Z) \ p_{S, \lambda}(Z \mid Y, T)$. Thanks a lot for pointing this out.
>
> [1] Causal Inference with Conditional Front-Door Adjustment and Identifiable Variational Autoencoder, ICLR 2024.

---

> > ### Comment · Reviewer_PTBp · 2024-11-19
> >
> > Thanks for the addressing my concerns. I have adjusted my score accordingly.

---

> > > ### Author Response · Authors · 2024-11-20
> > > **Thank You for the Acknowledgment**
> > >
> > > Dear reviewer PTBp,
> > >
> > > We sincerely appreciate your thoughtful feedback and thank you again for your time and efforts in the review and discussion phases.
> > >
> > > Best,
> > > Authors

---

### Official Review · Reviewer_7C4u · 2024-11-05

**Soundness:** 2
**Presentation:** 2
**Contribution:** 2
**Rating:** 5
**Confidence:** 3

**Summary:**

This paper studies the estimation of treatment effects under the presence of latent confounders and latent post-treatment variables. This is a challenging scenario where conventional methods may yield biased estimates of treatment effects. To correct the bias, this paper proposes a novel method, namely Confounder-identifiable VAE (CiVAE), to estimate treatment effects. The paper provides some theoretical guarantees for the method. Moreover, the paper demonstrates that the proposed method performs well in empirical applications.

**Strengths:**

- The problem is well-motivated.
- The method seems to work well in empirical applications.
- Figures 1 and 2 are very helpful for understanding the problem setup.

**Weaknesses:**

- The real-world dataset used in the empirical study does not seem to be publicly available.
- The paper uses many notations without definition. For example, the definitions of $K_Z$ in footnote 3, $p(X\mid C)$, and the matrix of $p(C, X)$ are missing.
- Section 4 is a bit challenging to follow -- notations are heavy, and the exposition is a bit detail-oriented. Providing some high-level intuitions in this section could be helpful.
- I am not sure why there is always a clear separation between latent confounders and post-treatment variables. It is possible that the value of latent confounders (e.g., health indicators) can be changed by the treatment, and thus, these latent confounders also seem to be post-treatment variables.

**Questions:**

See Weaknesses above.

---

> ### Author Response · Authors · 2024-11-18
> **Official Response of Submission5496 by Authors**
>
> Thank you very much for your constructive feedback. It's our pleasure to have this valuable opportunity to discuss your concerns/questions with you. We believe that our paper will be much stronger thanks to your efforts.
>
> > The real-world dataset used in the empirical study does not seem to be publicly available.
>
> Due to the company's policy, we are unable to share internal data during the review process. However, once the paper is made public, we will seek permission to make the dataset publicly available.
>
> >  The paper uses many notations without definition. For example, the definitions of  K_{Z}  in footnote 3,  p(X∣C), and the matrix of p(C, X)  are missing.
> >
> Thank you for pointing out the missing definitions. $K_{Z}$ was defined in Line 135 of Section 2 as $K_{Z} = K_{C} + K_{M}$, representing the dimension of the latent space. Additionally, we have added a definition for $p(X|C)$ as the conditional distribution of $X$ given $C$, and specified the matrix of $p(X|C)$ as $[p(X_{i}|C_{j})]^{K_{X}, K_{C}}_{i,j=1,1}$ in Section 3.1. Furthermore, we have thoroughly reviewed the paper to ensure that all notations are clearly defined before they are used.
>
> >  Section 4 is a bit challenging to follow -- notations are heavy, and the exposition is a bit detail-oriented. Providing some high-level intuitions in this section could be helpful.
>
> We apologize for the confusion caused by the presentation in Section 4. Based on your advice, we have made major revisions to improve clarity. We have added a high-level overview at the beginning of Section 4 to summarize the core concepts of CiVAE with reduced notations. Additionally, we have included high-level explanations at the start of subsections 4.1 to 4.4 to clarify the purpose of each component. We have also provided intuitive explanations and justifications following each theorem to enhance understanding. Please refer to the highlighted parts in the revised Section 4 for these improvements.
>
> >  I am not sure why there is always a clear separation between latent confounders and post-treatment variables. It is possible that the value of latent confounders (e.g., health indicators) can be changed by the treatment, and thus, these latent confounders also seem to be post-treatment variables.
>
> We appreciate your insightful question regarding the separation between latent confounders and post-treatment variables. In causal inference, it is crucial to consider the *time* when covariates are measured, although this is often omitted in the literature for simplicity. For example, health indicators measured before treatment are clear-cut pre-treatment variables. In contrast, if these indicators are measured after treatment, they can be divided into two subsets (in an ideal case for intuitive explanation): those that remain unchanged by the treatment, which are determined by pre-treatment health (i.e., a confounder), and those altered by the treatment, which are post-treatment variables.
>
> This scenario is common in observational studies when data used for causal effect estimation are collected *post hoc*, where the observed covariates are generated from an entanglement of latent confounders and latent post-treatment variables (see similar examples in social science, economics, and politics discussed in the Lines 73-101). The proposed CiVAE is the first work that provides provable identification and disentanglement of latent confounders from covariates with mixed-in latent post-treatment variables and empirically shows substantially improved robustness against latent post-treatment bias.

---

> ### Author Response · Authors · 2024-11-27
> **[Last day of paper revision] We are anticipating your feedback!**
>
> Dear reviewer 7C4u,
>
> It's the last day that we are allowed to revise the paper. We would be so grateful if you could kindly check our responses and revised paper and let us know if you are happy with the improvement we've made according to your advice.
>
> Please do not hesitate to reach out if you have any further questions or require additional clarifications. We are more than happy to have the valuable chance to continue the discussion with you.
>
> Thank you very much for your time and consideration.
>
> Sincerely,
> Authors

---

### Meta-Review · Area_Chair_DwE3 · 2024-12-21

**Metareview:**

The paper proposes a method to estimate treatment effects by disentangling latent confounders from latent post-treatment variables

Strengths:

+ Studies an important and understudied problem of post-treatment bias in causal inference


Weaknesses:

+ Relies on strong assumptions that are difficult to verify in practice, e.g. independence between latent confounders and post-treatment variables

+ Heavy notation, undefined terms, and lack of high-level intuition in the paper

**Additional Comments On Reviewer Discussion:**

The reviewers largely recognized the importance of the problem being studied, but had reservations on the practical relevance and the clarity of the presentation.

---

### Decision · Program_Chairs · 2025-01-22

Accept (Poster)